# Sparse Autoencoders are Topic Models

**Leander Girrbach** [1]  **Zeynep Akata** [1]

## Abstract

Sparse autoencoders (SAEs) are used to analyze embeddings, but their role and practical value are debated. We propose a new perspective on SAEs by demonstrating that they can be naturally understood as topic models. We propose a continuous topic model (CTM) inspired by Latent Dirichlet Allocation (LDA) for embedding spaces and derive the SAE objective as a maximum a posteriori estimator under this model. This view implies SAE features are thematic components rather than steerable directions. To confirm our theoretical findings, we introduce SAE-TM, a topic modeling framework that: (1) trains an SAE to learn reusable topic atoms, (2) interprets them as word distributions on downstream data, and (3) merges them into any number of topics without retraining. SAE-TM yields more coherent topics than strong baselines on text and image datasets while maintaining diversity. Finally, we analyze thematic structure in image datasets and trace topic changes over time in Japanese woodblock prints. Our work positions SAEs as effective tools for large-scale thematic analysis across modalities. Code is available at ExplainableML/SAE-TM.

## 1. Introduction

Sparse autoencoders (SAEs) are an important tool for understanding embedding spaces, particularly the internal activations of foundation models (Bricken et al., 2023; Kim et al., 2025). However, practical, high-impact applications have remained limited, and SAEs have been criticized because of failures in steering (Wu et al., 2025; Peng et al., 2025) and for being inferior to linear probes (Smith et al., 2025). This raises important questions: (i) how should we understand SAEs, and (ii) how can we best use their strengths?

[1]Technical University of Munich (TUM), Munich Center for Machine Learning (MCML), Helmholtz Munich. Correspondence to: Leander Girrbach <leander.girrbach@tum.de>.

*Proceedings of the 43rd International Conference on Machine Learning*, Seoul, South Korea. PMLR 306, 2026. Copyright 2026 by the author(s).

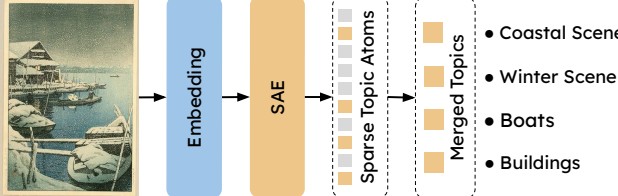

*Figure 1.* We show theoretically and practically that Sparse Autoencoders are strong topic models in embedding space. An embedding model converts documents, such as images or text, into embeddings. An SAE then encodes these to produce sparse topic atoms. These atoms are merged into coarse-grained topics that are relevant to the entire dataset. Finally, we show that SAE Topic Models can be successfully used to analyze large-scale data.

In this paper, we argue that SAEs are naturally understood as topic models, i.e. models that represent each datapoint as a mixture of prominent themes found in the entire dataset. Concretely, we propose a continuous topic model (CTM) inspired by Latent Dirichlet Allocation (Blei et al., 2003, LDA) for embedding spaces and derive the SAE objective as a MAP estimator under this model. This implies that SAE features should be seen as thematic clusters whose activations combine to explain an embedding, rather than a monosemantic, steerable mechanism. Consequently, SAEs may indeed be less suited for mechanistic control at the level of single features. Instead, they work well for discovering and organizing unknown themes in data. We operationalize this view by constructing topic models directly from SAEs: we pretrain once to learn reusable topic atoms, interpret them as word distributions on downstream datasets, and merge them into any desired number of topics. We then evaluate the resulting models against strong baselines in both text and image settings.

In this way, SAE topic models also enable large-scale thematic analysis of image datasets. Although such datasets are central to computer vision research and applications, their broader thematic content is underexplored. By representing images as mixtures of learned topic atoms, SAEs enable efficient inspection of recurring visual themes within and across datasets. This offers a new perspective on dataset differences and similarities. We apply our approach to four widely used image datasets and find systematic and interpretable contrasts in their thematic structure.

In summary, our contributions are as follows: (1) we formalize a connection between topic models and SAEs by intro-

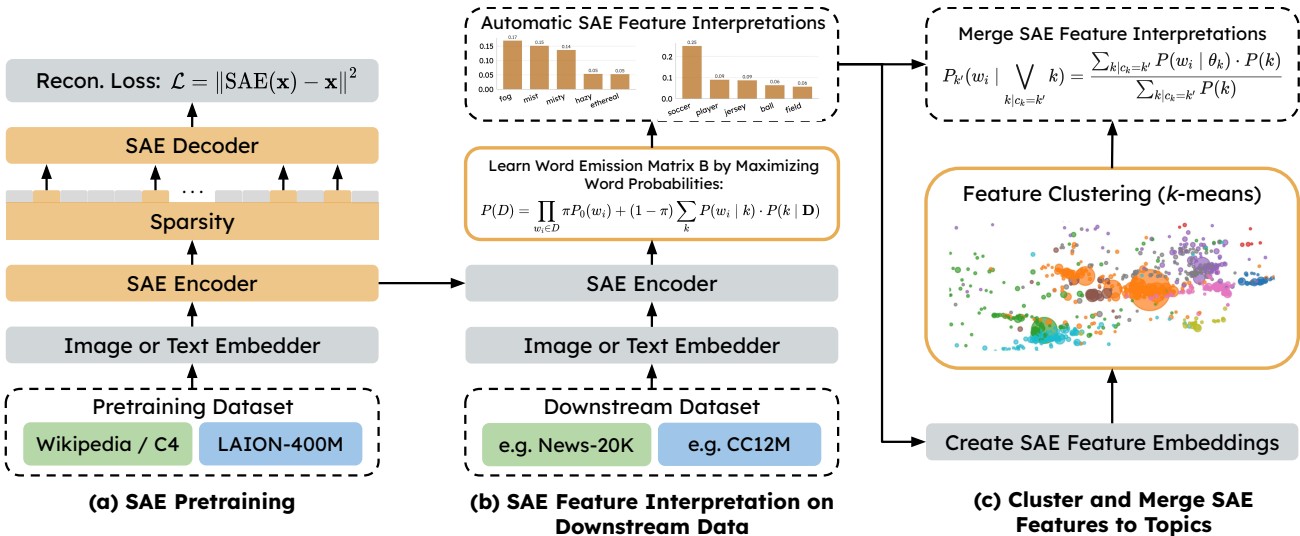

*Figure 2.* Overview of our SAE topic model (SAE-TM): (a) pretrain foundational SAEs on large text or vision datasets to learn transferable atomic directions; (b) interpret relevant SAE features on downstream datasets by associating each feature with a distribution over words; (c) cluster SAE feature embeddings derived from their top associated words via $k$-means and merge clustered features into topics, aggregating their word distributions. Colors indicate modality (green = text, blue = vision) and trainable (orange) vs. frozen (grey) components.

ducing a continuous topic model inspired by LDA and deriving the SAE objective as a MAP estimator under this model; (2) we show how to use SAEs as foundational topic models and find that they compare favorably to strong baselines on standard coherence and diversity metrics; (3) we apply our SAE-based topic model to analyze differences in the composition of four popular large-scale image datasets, finding clear and interpretable differences in their thematic structure, such as object-centric vs. human-centric emphases; and (4) we demonstrate how our SAE-TMs can detect changes in themes in Japanese woodblock prints across periods. An overview is in Figure 2.

## 2. Related Work

**Topic Modeling.** Most topic models infer topics as latent variables by maximizing the likelihood of the data. Early examples are LDA (Blei et al., 2003) and pLSI (Hofmann, 1999). These have inspired many extensions, including leveraging document information (Blei & Lafferty, 2005), dynamic topics (Blei & Lafferty, 2006), and minibatch training (Hoffman et al., 2010). In Neural Topic Models (NTMs), Srivastava & Sutton (2017) combine LDA and VAEs (Kingma & Welling, 2014), replacing the Dirichlet prior of LDA with a logistic normal distribution, which has become a standard approach of NTMs (Miao et al., 2016; 2017; Card et al., 2018; Burkhardt & Kramer, 2019; Dieng et al., 2020). Other works improve topic quality, e.g., using a Wasserstein autoencoder (Nan et al., 2019), a Weibull VAE to mitigate Gaussian latent problems (Zhang et al., 2018), extend to multimodal data (Abaskohi et al., 2025),

or increase sparsity (Lin et al., 2019; Martins & Astudillo, 2016). Beyond VAEs, optimal transport infers word-topic mappings via transport plans between embeddings (Zhao et al., 2021; Guo et al., 2022; Wu et al., 2023; 2024b). Other methods use contrastive learning (Nguyen & Luu, 2021) or prompt LLMs (Pham et al., 2024). Clustering document embeddings is also popular, where each cluster forms a topic (Aharoni & Goldberg, 2020; Sia et al., 2020; Angelov, 2020; Thompson & Mimno, 2020; Grootendorst, 2022; Zhang et al., 2022). The main weaknesses of current NTMs are suffering from posterior collapse (Bowman et al., 2016; Subramanian et al., 2018), having an inflexible number of topics, and being heavily tailored towards text analysis. Our SAE-TM directly addresses those weaknesses.

**Sparse Autoencoders.** Sparse coding learns an overcomplete basis, using only a fraction of basis vectors per data point. Sparse Autoencoders (SAEs) have been shown to learn expressive, disentangled features (Olshausen & Field, 1996; Ranzato et al., 2006; 2007; Makhzani & Frey, 2014). The ability of SAEs to learn sparse, independent directions enables LLM activation interpretation (Cunningham et al., 2023; Bricken et al., 2023) by extracting "monosemantic" directions (Elhage et al., 2022; Klindt et al., 2025). We leverage this property to extract topic atoms from superimposed topics in document embeddings. Recent SAE improvements target expressivity (Rajamanoharan et al., 2024; Nabeshima, 2024), sparsity (Bussmann et al., 2024; Gao et al., 2025), separability (Hindupur et al., 2025; Engels et al., 2025a;b), and feature hierarchies (Fel et al., 2025; Muchane et al., 2025). Recent work (Jiang et al., 2025; Choi et al., 2025) also proposes using SAEs for dataset inspection, but does

| (a) Classical LDA (discrete) | (b) Continuous Topic Model (embeddings) |
| --- | --- |
| **Hyperparams**: number of topics $K$; Dirichlet $\alpha \in \mathbb{R}^K_{>0}$; word dists $\beta \in [0,1]^{K \times V}$ with rows on the simplex; doc-length rate $\xi$. | **Hyperparams**: $K$; Dirichlet $\alpha \in \mathbb{R}^K_{>0}$; directions $\mu_{1:K} \in \mathbb{R}^d$; covariances $\Sigma_{1:K} \succeq 0$; strength dists $\mathrm{Ga}_{1:K}$ on $\mathbb{R}_{\geq 0}$; Poisson rate $\rho_d$; noise $\sigma^2$. |
| **1. Topic mix**: $\theta \sim \mathrm{Dir}(\alpha)$. | **1. Topic mix**: $\theta \sim \mathrm{Dir}(\alpha)$. |
| **2. Length**: $N \sim \mathrm{Pois}(\xi)$. | **2. # contributions**: $N \sim \mathrm{Pois}(\rho_d)$. |
| **3. For** $n = 1{:}N$: 
 (a) topic $z_n \sim \mathrm{Cat}(\theta)$; 
 (b) word $w_n \sim \mathrm{Cat}(\beta_{z_n})$. | **3. For** $n = 1{:}N$: 
 (a) topic $z_n \sim \mathrm{Cat}(\theta)$; 
 (b) direction $w_n \sim \mathcal{N}(\mu_{z_n}, \Sigma_{z_n})$; 
 (c) strength $\lambda_n \sim \mathrm{Ga}_{z_n}$; 
 (d) contribution $c_n = \lambda_n w_n$. |
| **4. Document (obs.)**: bag-of-words counts $X = \sum_{n=1}^N e_{w_n}$. 
 **Mean (given $\theta$)**: $\mathbb{E}[X \mid \theta] = \beta^\top \theta$. | **4. Embedding (obs.)**: $D = \sum_{n=1}^N c_n + \varepsilon$, with $\varepsilon \sim \mathcal{N}(0, \sigma^2 I)$. 
 **Mean (given $\theta$)**: $\mathbb{E}[D \mid \theta] = W\theta$, where 
 $W = [\rho_d m_1 \mu_1, \ldots, \rho_d m_K \mu_K]$, $m_k = \mathbb{E}[\lambda \mid z{=}k]$. |

*Table 1.* Side-by-side comparison of the generative processes for classical LDA (discrete) and the proposed continuous topic model (CTM) for embedding spaces inspired by LDA. Steps are aligned to highlight both the shared structure and the differences.

not make a link to topic modeling. Zheng et al. (2025) adapt existing topic modeling methods to use SAE features as input tokens, while we theoretically establish that the SAE objective itself is a MAP estimator of a topic model.

## 3. Comparing SAEs and Topic Models

### 3.1. Background on Latent Dirichlet Allocation

Latent Dirichlet Allocation (LDA) is a generative probabilistic model for collections of discrete data such as text corpora (Blei et al., 2003). Each document is represented as a mixture of latent topics, where each topic is a distribution over words. Formally, for a document $w = (w_1, \ldots, w_N)$ in a corpus with vocabulary size $V$ and $K$ topics, the process is:

- Draw topic proportions $\theta \sim \mathrm{Dir}(\alpha)$.
- For each position $n = 1, \ldots, N$ (with $N \sim \mathrm{Pois}(\xi)$):
  - Sample topic $z_n \sim \mathrm{Cat}(\theta)$.
  - Sample word $w_n \sim \mathrm{Cat}(\beta_{z_n})$, where $\beta$ is a $K \times V$ matrix of word distributions.

The joint distribution is $p(\theta, z, w | \alpha, \beta) = p(\theta|\alpha) \prod_{n=1}^N p(z_n|\theta) p(w_n|z_n, \beta)$, and integrating over $\theta$ yields the marginal $p(w|\alpha, \beta)$. Thus LDA consists of corpus-level parameters $(\alpha, \beta)$, document-level topic proportions $(\theta_d)$, and word-level assignments $(z_{dn}, w_{dn})$.

### 3.2. A Continuous Topic Model Inspired by LDA

We formalize a Continuous Topic Model (CTM) analogous to LDA, but operating on document embeddings $D \in \mathbb{R}^d$ instead of words. This makes topic modeling possible for other domains, such as vision, in addition to language, where similar precedents exist that learn topic models on embedding spaces (Dieng et al., 2020; Das et al., 2015). Our core assumption is that each document embedding is a linear

combination of topic-specific continuous directions, which is a direct instantiation of the linear representation hypothesis (Park et al., 2024), i.e., embeddings are linear mixtures of factors. Formally, we represent $D$ as $D = \epsilon + \sum_{i=1}^N \lambda_i c_i$, where $c_i$ are the directions, $\lambda_i$ are strength coefficients that determine the scale of each direction, and $\epsilon$ is the variation left unexplained by the directions.

This formulation adapts LDA's structure for embeddings: The $N$ contributions correspond to the number of words (or components in an abstract sense) in a document. For each contribution, we sample a single topic $z_n \sim \mathrm{Cat}(\theta)$ from a document-level categorical distribution, where $\theta \sim \mathrm{Dir}(\alpha)$. As in LDA, $\alpha$ is a corpus-level parameter, and each contribution is assumed to be generated from one topic.

Finally, instead of sampling a discrete word from a categorical distribution, we sample a scaled continuous topic vector $\lambda_n w_n$. We independently sample the strength $\lambda_n$ from a Gamma distribution $\mathrm{Ga}_{z_n}$ and the continuous direction $w_n$ from a Gaussian distribution $\mathcal{N}(\mu_{z_n}, \Sigma_{z_n})$. However, note that $\mu_k$ are topic directions in document-embedding space. They are not embeddings of vocabulary items, and the CTM does not include topic-word distributions. A comparison of LDA and our CTM is in Table 1.

The expected document is linear in $\theta$, $\mathbb{E}[D|\theta] = W\theta$ with columns $W_{\cdot k} \propto \mu_k$, directly paralleling $\mathbb{E}[w|\theta] = \beta^\top \theta$ in LDA. Thus, both models share the same linear mixture structure, but differ in observation models (multinomial vs. Gaussian) and the data domain (discrete vs. continuous).

The main difference between the CTM and LDA is the introduction of the strength $\lambda$. This parameter is necessary to cover the entire space $\mathbb{R}^d$ with a finite mixture of topic directions. It is also motivated by viewing distributions of $w_n \sim \mathcal{N}(\mu_k, \Sigma_k)$ as directions instead of centroids.

### 3.3. Relating SAEs with $L_1$ Penalty to the CTM

In this section, we show that the SAE objective with $L_1$ penalty (Bricken et al., 2023) arises as a MAP estimator under the CTM of Section 3.2 with the following assumptions:

**(A1)** *High-activity, small-contribution limit:* $\rho_d \to \infty$ and $\alpha_0 \to 0$, with $\rho_d \alpha_0 \to \kappa \in (0, \infty)$, so that many small-strength contributions accumulate to a finite total.

**(A2)** *Concentrated topic directions:* $\Sigma_k \to 0$, so each contribution $w_n$ aligns with the topic mean direction $\mu_{z_n}$.

**(A3)** *Independence across topics:* the per-topic aggregated strengths $S_k$ are mutually independent given $\theta$.

Under (A1)–(A3), $L_1$-penalty SAEs arise as the MAP solution to the CTM in the high-activity, small-contribution limit, as we now derive. Fixed-sparsity SAEs such as TopK (Gao et al., 2025) and BatchTopK (Bussmann et al., 2024) admit an analogous derivation under (A2)–(A3) that replaces (A1) with a hard support constraint (see Appendix A).

Assume the strength of each continuous contribution follows a Gamma distribution with common rate $\beta > 0$ and shape $\alpha_0 > 0$ across topics,

$$\lambda_{k,i} \sim \mathrm{Ga}(\alpha_0, \beta), \qquad k = 1, \ldots, K. \tag{1}$$

Let $N \to \infty$ while each individual contribution becomes small. Concretely, we consider the limit

$$\rho_d \to \infty, \quad \alpha_0 \to 0, \quad \text{with } \rho_d \alpha_0 \to \kappa \in (0, \infty), \tag{2}$$

so many small-strength contributions accumulate to a finite total. Writing $N_k \sim \mathrm{Pois}(\rho_d \theta_k)$ for the number of contributions to topic $k$, the aggregated strength for topic $k$,

$$S_k := \sum_{i=1}^{N_k} \lambda_{k,i}, \tag{3}$$

converges in distribution to

$$S_k \Rightarrow \mathrm{Ga}\left(\kappa \theta_k, \beta\right). \tag{4}$$

Define the total strength and normalized topic weights

$$s := \sum_{k=1}^{K} S_k, \qquad \tilde{\theta}_k := \frac{S_k}{s}. \tag{5}$$

By independence across topics, we obtain

$$s \Rightarrow \mathrm{Ga}(\kappa, \beta), \qquad \tilde{\theta} \Rightarrow \mathrm{Dirichlet}(\kappa\theta), \tag{6}$$

and $(s, \tilde{\theta}) \perp\!\!\!\perp$ given $\theta$. When $\kappa$ is large (i.e., many small contributions), $\tilde{\theta}$ concentrates on the document-level mixture $\theta$, and we may work with the collapsed representation

$$s \sim \mathrm{Ga}(\kappa, \beta), \quad \theta \sim \mathrm{Dir}(\alpha), \quad D = sW\theta + \varepsilon, \tag{7}$$

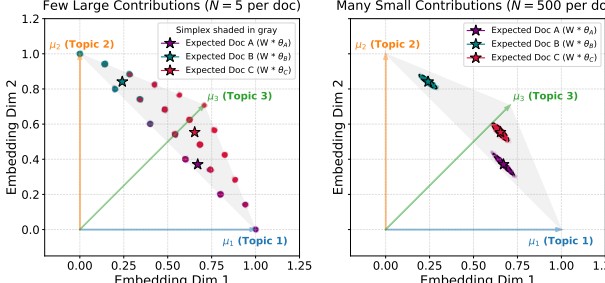

*Figure 3.* Illustration of the high-activity, small-contribution limit (A1). We sample document embeddings from the CTM (Section 3.2) for three documents $A$, $B$, $C$ with distinct topic mixtures $\theta_A, \theta_B, \theta_C$ over three topic directions $\mu_1, \mu_2, \mu_3$ (colored arrows; the topic simplex is shaded gray). Stars mark each document's expected embedding $W\theta$. **Left** ($N = 5$, few large contributions): samples form a discrete, clumpy pattern that is poorly matched to the continuous $L_2$ reconstruction loss used by SAEs. **Right** ($N = 500$, many small contributions): samples concentrate into Gaussian clouds around $W\theta$, recovering the SAE observation model of Equation (7).

with $\varepsilon \sim \mathcal{N}(0, \sigma^2 I)$ and $W = [\mu_1, \ldots, \mu_K] \in \mathbb{R}^{d \times K}$, corresponding to the limit $\Sigma_k \to 0$ so that contributions align with $\mu_k$. Reparameterizing by

$$a_k := s, \ \theta_k \Leftrightarrow s = \sum_{k=1}^{K} a_k = \|a\|_1, \ \theta_k = \frac{a_k}{\sum_j a_j}, \tag{8}$$

the observation model in Equation (7) becomes the standard SAE decoder $D \mid a \sim \mathcal{N}(Wa, \sigma^2 I)$ with $a \geq 0$. With the Gamma prior $s \sim \mathrm{Ga}(\kappa, \beta)$ and Dirichlet prior $\theta \sim \mathrm{Dir}(\alpha)$, the negative log-posterior for a single embedding $D$ reads

$$\mathcal{L}(\theta, s) = \frac{1}{2\sigma^2}\|D - Wa\|_2^2 + \underbrace{\beta s + (1 - \kappa)\log s}_{-\log p(s)}$$
$$+ \underbrace{\sum_{k=1}^{K}(1 - \alpha_k)\log\theta_k}_{-\log p(\theta)} + \text{const}, \qquad a = s\theta. \tag{9}$$

Choosing $\kappa = 1$ and $\alpha_k = 1$ yields an exponential prior on $s$ and uniform Dirichlet on $\theta$, giving the SAE objective with $L_1$ penalty (Bricken et al., 2023)

$$\mathcal{L}(a) = \frac{1}{2\sigma^2}\|D - Wa\|_2^2 + \beta\|a\|_1 + \text{const}, \quad a \geq 0. \tag{10}$$

When $\kappa < 1$, the additional $(1 - \kappa)\log s$ term further encourages a smaller total mass $s$, while $\alpha_k < 1$ promotes peaked usage within the active topics.

**Illustration.** To illustrate the theory between how SAEs constrain the high-activity, small-contribution limit, we show in Figure 3 how embeddings for 3 documents are generated under different conditions. On the left, a document embedding

is formed by only a few large topic contributions ($N = 5$). In this case, possible embeddings form a rigid, clumpy grid. This discrete pattern doesn't harmonize well with the continuous $L_2$ reconstruction loss used by SAEs. However, on the right, we move to the high-activity, small-contribution limit ($N = 500$). There, the clumpy grid smoothes out into Gaussian clouds centered at each document's expected embedding ($W\theta$).

This explains how we can apply the standard SAE objective (i.e. the continuous $L_2$ loss) to a model based on discrete, word-like topic contributions (the CTM). At the same time, making each individual contribution small prevents the overall embedding size from increasing without bound, which mathematically yields the $L_1$ sparsity regularization.

## 4. Applying SAEs as Topic Models

Section 3 establishes that the SAE objective is a MAP estimator under the CTM: in the latent-variable sense, the SAE itself is the topic model, with each feature acting as a topic atom and its activation as the inferred per-document topic weight. To evaluate this topic model against standard baselines and to interpret its topics, however, two practical gaps must be bridged. First, SAEs typically have $\gg 1000$ features, while traditional topic models use a much smaller number of topics. Second, topics are commonly defined as a distribution over words (Srivastava & Sutton, 2017; Wu et al., 2024b), an interpretation that SAEs do not directly support. We therefore introduce two post-hoc steps on top of the frozen SAE: (i) learning a word-emission matrix so that SAE features can be inspected as word distributions and evaluated with standard topic-model metrics, and (ii) merging features into a smaller set of topics to match the granularity used by baselines. Both steps operate on a frozen SAE and do not modify the generative model of Section 3, so they are interpretation layers, not a new topic model.

**SAE topic interpretation.** We consider a trained SAE with a large number of latent features, $K \gg 1,000$. Our goal is to interpret these SAE features as topics, where each topic is a distribution over words, following conventions in topic modeling as noted above. To interpret SAE features, we learn the word emission matrix $\mathbf{B} \in \mathbb{R}^{K \times V}$ by maximizing the bag-of-words likelihood of each datapoint given its SAE feature probabilities:

$$P(D) = \prod_{w_i \in D} \pi P_0(w_i) + (1-\pi) \sum_k \underbrace{P(w_i \mid k)}_{=B_{k,i}} \cdot \underbrace{P(k \mid \mathbf{D})}_{=\theta_k}, \tag{11}$$

where $D$ is a textual representation of the document (for example, a detailed caption of an image), $\mathbf{D}$ is its embedding, and $P(\theta_k \mid \mathbf{D})$ is the normalized activation of the $k$-th SAE feature, as defined in Equation (8). The background unigram prior $P_0(w)$ is an unconditional prior probability

over words that accounts for words that are generally frequent but have no specific topic associations. This prevents the SAE feature interpretations from needing to model such common words. $\pi$ is set to $0.3$ in all experiments (see Appendix E for a systematic ablation). Additionally, we also improve SAE feature interpretations by weighting the contribution of each word to the document-level loss by its normalized inverse-document weight $\frac{\log \frac{N}{\mathrm{df}(w_i)}}{\max_j \log \frac{N}{\mathrm{df}(w_j)}}$, where $N$ is the total number of documents and $\mathrm{df}(w_i)$ is the number of documents where word $w_i$ appears. We learn $\mathbf{B}$ by minimizing $-\log P(D)$ over the entire dataset.

**Topic merging.** As noted, SAEs typically have many features ($\gg 1,000$), while topic modeling requires fewer topics for better interpretability. We therefore treat SAE features as *topic atoms* and construct broader topics from them. This approach has key advantages. The number of topics can be chosen flexibly using validation metrics without requiring retraining of the model, making this approach computationally effective. Additionally, we can directly assess the distinctness of the topics and use this information to decide which ones to merge.

Given $K$ SAE features and a target number of topics $K' < K$, we merge topics by creating embeddings for the SAE features and clustering them into $K'$ groups using $k$-means clustering. Topic embeddings are constructed as the weighted sum of word embeddings for the $V$ words in the vocabulary $\mathcal{V}$ used for feature interpretation. The topic embedding $\mathbf{T}_k$ is then defined as $\mathbf{T}_k = \sum_{w_i \in \mathcal{V}} B_{k,i} \cdot \mathbf{w}_i$, where $\mathbf{w}_i$ is the embedding of the word $w_i$. Additionally, we find it useful to denoise the topic embeddings by only considering the top-$p$ vocabulary (Holtzman et al., 2020), i.e. taking the smallest set of words whose cumulative probability exceeds $p$ (we use $p = 0.9$) and renormalizing so probabilities sum to 1.

For embeddings, we use widely available models like word2vec (Mikolov et al., 2013) or GloVe (Pennington et al., 2014). Alternatively, if word embeddings are not available, the SAE decoder weights for each feature also provide good topic embeddings (Rao et al., 2024). After obtaining a cluster label $c_k$ for each topic, we merge the corresponding rows in $\mathbf{B}$ as follows:

$$P_{k'}(w_i \mid \bigvee_{k|c_k=k'} k) = \frac{\sum_{k|c_k=k'} P(w_i \mid \theta_k) \cdot P(k)}{\sum_{k|c_k=k'} P(k)}, \tag{12}$$

where $P(k)$ is the average $\theta_k$ across all datapoints. Appendix C contains an extended pseudocode and implementation details for the reader's convenience.

**Foundational SAE topic models.** Dynamically creating topics from topic atoms is also highly advantageous when working with limited data, similar to clustering pretrained embeddings. When building a topic model for a smaller

dataset that cannot support training an expressive SAE from scratch, we can reuse the topic atoms from the pretrained SAE. We then use the statistics of the small dataset to select the relevant atoms and decide how to merge them.

# 5. Evaluating SAE Topic Models

**Baselines.** We compare SAEs as topic models against representative state-of-the-art neural topic models: AVITM (Srivastava & Sutton, 2017), CombinedTM (Bianchi et al., 2021a;b), DecTM (Wu et al., 2021), DVAE (Burkhardt & Kramer, 2019), ETM (Dieng et al., 2020), FASTopic (Wu et al., 2024b), NSTM (Zhao et al., 2021), and TSCTM (Wu et al., 2022). All implementations except DVAE are adapted from TopMost (Wu et al., 2024c). Among these models, only CombinedTM and FASTopic operate on embeddings (like SAEs). All other baselines require bag-of-words representations of documents as inputs. This highlights the necessity to develop embedding-based topic models for image applications. We do not include BERTopic (Grootendorst, 2022) because it targets a different definition of topic modeling: documents are clustered in embedding space, and each is assigned at most one topic. This constrasts to other baselines and also to SAE-TM, which assign multiple topics to each document, so results are not comparable.

**SAE Training.** All SAEs are trained and interpreted on individual datasets to match baselines (note that in Figure 2 we adopt the "foundational SAE" perspective introduced above, but train SAEs on individual datasets here for fair comparison to baselines). We use the following hyperparameters: Expansion factor 4 (i.e. the dictionary size is 3072 for all models), $L_1$ penalty is 2, batch size is 1000, and we train SAEs for 50,000 steps with learning rate 0.001. For interpretation, we use a vocabulary size 5000 (same for all baselines), batch size 1000, and learning rate 0.01. The number of epochs depends on the dataset (min. 50, max. 200). Training and interpretation are highly efficient, e.g. training an SAE on 50M embeddings (Twitter dataset) takes 10 minutes on a single GPU, interpretation 15 minutes.

**Evaluation Metrics.** Topic model evaluation is known to be challenging (Chang et al., 2009; Lau et al., 2014; Hoyle et al., 2021; Harrando et al., 2021; Rahimi et al., 2024). The most important evaluation axes are *Topic Coherence* and *Topic Diversity* (Wu et al., 2024b;a). Topic Coherence measures topic clarity and specificity, aiding interpretation. Topic Diversity measures inter-topic overlap and controls for models manipulating coherence by learning and repeating a few narrow topics.

We measure topic coherence using two metrics: Overall topic rating $C_R$ (Newman et al., 2010) and intruder detection $C_I$ (Chang et al., 2009). Overall rating scores the relatedness of a topic's top 20 words on a scale from 0 to 100. For intruder detection, we repeatedly sample 5 of the top-20 words from a topic and add an intruder word from a different topic. A judge must detect the intruder, and the score is the judge's accuracy. Following Stammbach et al. (2023); Rahimi et al. (2024), we use LLMs as judges (PHI-4; Abdin et al. (2024)), as prior work has established they align with human judges. Coherence scores are averages across topics.

We measure topic diversity by the avg. word mover distance (Kusner et al., 2015, WMD) between topics. For topics $k, k'$ represented by their top 20 words, WMD is defined as

$$\text{WMD}(k, k') = \min_{T \geq 0} \sum_{i,j} T_{i,j} C_{i,j}, \quad (13)$$

where $C_{i,j} = \|\mathbf{w}_i - \mathbf{w}'_j\|_2$ is the distance of paired word embeddings and $T$ follows optimal transport constraints (uniform marginals). Intuitively, WMD calculates the "cost" to "move" all word embeddings from one topic to the other. WMD ranks models similarly to the ratio of unique words (Dieng et al., 2020), but is robust to varying topic numbers (see Appendix D).

Finally, we also demonstrate that SAE-TM does not only learn coherent and diverse topics, but topics are relevant to the respective documents. For this, we sample one active topic and one inactive topic per document and let an LLM decide if the topic is relevant to the document. See Appendix F for the results on this metric.

## 5.1. Evaluation on Text-Only Datasets

**Datasets.** We evaluate topic models on five text-only datasets: News-20K (Mitchell, 1997), IMDB movie reviews (Maas et al., 2011), Yelp restaurant reviews (Zhang et al., 2015), DailyMail stories (Hermann et al., 2015), and a filtered tweet dataset (Cheng et al., 2010). These datasets cover diverse domains such as reviews (IMDB and Yelp), news (News-20K and DailyMail), and social media (Twitter). They also vary in size: News-20K has 18,846 documents, IMDB and Yelp each have 50,000, DailyMail has 219,507 articles, and the filtered Twitter dataset has 1,183,728 tweets. The original Twitter dataset contains over three million tweets, but we only keep those with at least seven non-stopword lemmas from the top 5,000 most common lemmas to ensure sufficient content for topic detection. We preprocess documents as follows: We lemmatize all words and filter stopwords. In each dataset, we determine the 5,000 most frequent lemmas and ignore all others. We use the NLTK library (Bird, 2006) for tokenization, lemmatization, and stopword filtering. For document embeddings, we use GRANITE-R2 (Awasthy et al., 2025).

**Results.** Table 2 shows metrics averaged across text-only datasets, but for different numbers of topics. SAE-TM outperforms all baselines in topic coherence, achieving the

| Num. Topics | 50 | | | 100 | | | 200 | | | 300 | | | 500 | | |
|---|---|---|---|---|---|---|---|---|---|---|---|---|---|---|---|
| | $C_I \uparrow$ | $C_R \uparrow$ | $D \uparrow$ | $C_I \uparrow$ | $C_R \uparrow$ | $D \uparrow$ | $C_I \uparrow$ | $C_R \uparrow$ | $D \uparrow$ | $C_I \uparrow$ | $C_R \uparrow$ | $D \uparrow$ | $C_I \uparrow$ | $C_R \uparrow$ | $D \uparrow$ |
| AVITM (Srivastava & Sutton, 2017) | 38.72 | 69.05 | 3.36 | 35.74 | 69.02 | 3.36 | 34.59 | 68.17 | 3.27 | 33.38 | 65.67 | 3.27 | 31.08 | 59.61 | 3.13 |
| CombinedTM (Bianchi et al., 2021a) | 40.90 | 70.24 | 3.24 | 38.49 | 67.37 | 3.22 | 37.56 | 63.55 | 3.45 | 27.78 | 42.72 | 3.21 | 31.79 | 50.77 | 3.24 |
| DecTM (Wu et al., 2021) | 38.85 | 66.49 | 3.26 | 35.43 | 66.00 | 3.21 | 28.93 | 53.14 | 3.15 | 25.22 | 42.18 | 3.11 | 20.37 | 40.34 | 2.98 |
| DVAE (Burkhardt & Kramer, 2019) | 21.24 | 22.45 | 3.00 | 17.36 | 11.16 | 3.36 | 16.87 | 2.29 | 3.21 | 16.64 | 4.69 | 3.20 | 16.50 | 2.87 | 3.17 |
| ETM (Dieng et al., 2020) | 21.68 | 28.36 | 3.08 | 20.26 | 23.28 | 3.18 | 20.02 | 19.05 | 3.25 | 19.77 | 17.91 | 3.29 | 19.23 | 15.62 | 3.34 |
| FASTopic (Wu et al., 2024b) | 32.31 | 56.06 | 2.92 | 30.33 | 56.90 | 2.96 | 29.46 | 51.83 | 2.89 | 28.24 | 52.34 | 2.97 | 28.06 | 51.25 | 2.97 |
| NSTM (Zhao et al., 2021) | 21.73 | 39.62 | 3.07 | 22.88 | 38.95 | 3.04 | 22.61 | 41.69 | 2.99 | 22.59 | 42.49 | 2.95 | 23.43 | 48.58 | 2.76 |
| TSCTM (Wu et al., 2022) | 44.61 | 69.75 | **3.87** | 35.81 | 58.53 | **3.79** | 29.51 | 40.00 | **3.76** | 26.17 | 27.40 | **3.70** | 21.68 | 17.67 | **3.70** |
| SAE-TM (ours) | **54.31** | **77.25** | 3.67 | **51.48** | **78.01** | 3.64 | **46.63** | **75.71** | 3.60 | **43.50** | **74.22** | 3.59 | **40.49** | **71.22** | 3.57 |

*Table 2.* Results for topic modeling performance on five text datasets. Values show topic coherence ($C_I$ = intruder detection accuracy, $C_R$ = topic coherence rating) and diversity ($D$) scores. All scores are averaged over datasets. Best values are in bold, and second-best values are underlined. Different numbers of topics show trends when increasing topic granularity.

highest scores for both intruder detection ($C_I$) and overall rating ($C_R$). Regarding topic diversity ($D$), SAE-TM consistently ranks second, trailing only TSCTM, which boosts diversity by upweighting semantically related, but low-frequency words. Additionally, the coherence of topics identified by TSCTM declines sharply as the number of topics increases, dropping from 69.75 ($C_R$) at 50 topics to 17.67 at 500 topics. SAE-TM, in contrast, maintains high and stable coherence scores even at 500 topics. The next-best performing baselines for coherence, typically AVITM and CombinedTM, still fall significantly short of the performance achieved by our SAE-TM.

### 5.2. Evaluation on Image Datasets

**Datasets.** We evaluate topic models on three image datasets: CIFAR100 (Krizhevsky et al., 2009) (50,000 images), Food101 (Bossard et al., 2014) (75,750 images), and SUN397 (Xiao et al., 2010) (108,618 images). To embed images, we use VIT-B-16-SIGLIP from OpenCLIP (Ilharco et al., 2021). We also create detailed, long-form captions by INTERNVL3.5-14B (Wang et al., 2025) for all images, which are necessary for learning topic models. For baselines that only process text data, we make a best-effort comparison by training them on captions. However, SAEs, FASTopic, and CombinedTM directly use image embeddings. Caption preprocessing for learning word emission probabilities follows Section 5.1. SAEs furthermore support alternative interpretation methods for images, such as top-$k$ most activating images per feature with subsequent description or keyword extraction, however we use only captions here to allow a fair comparison to baselines.

**Results.** Table 3 shows metrics averaged across three image datasets. Similar to text data, SAE-TM identifies significantly more coherent topics than other methods, which confirms its strong performance in analyzing image data as well. However, its topic diversity is weaker than some baselines but comparable to its performance on text datasets. This means other methods can increase the diversity of their topics on image datasets, while SAE-TM remains stable. We attribute this to modality effects, as image embeddings

focus on a few foreground objects (Li et al., 2025; Xiao et al., 2025; Kar et al., 2024), and SAE features sometimes represent concepts that cannot be explained well in language, subsequently binding high-frequency words in interpretation. However, this performance gap warrants further improvements to SAE feature interpretation to focus on topic-relevant words. As an orthogonal contribution, we observe that other methods that operate on image embeddings (CombinedTM and FASTopic) also yield good topics. CombinedTM performs well for a low number of topics, whereas FASTopic remains stable across different numbers of topics. This confirms the potential for learning TMs using image embeddings alone.

## 6. Inspecting Image Datasets with SAE TMs

### 6.1. Topic Distribution of Popular Datasets

We analyze the most prevalent topics and topic differences in popular image datasets, i.e. ImageNet (Deng et al., 2009), CC3M (Sharma et al., 2018), CC12M (Changpinyo et al., 2021), and YFCC-15M (Thomee et al., 2016). These datasets are widely used, but their content remains underexplored beyond concept frequency statistics (Garcia et al., 2023; Udandarao et al., 2024; Wiedemer et al., 2024). Hence, we utilize SAE topic models to examine diverging themes in these datasets and understand their differences.

**Dataset Preprocessing.** The combined datasets contain > 30 million images. To interpret the SAE topic model's features, we pair each image with a caption by INTERNVL3.5-14B. We follow the caption processing described in Section 5.1 and embed images with VIT-B-16-SIGLIP, and use the foundational SAE described in Appendix B.

**Deriving topics.** We learn word emission probabilities for SAE features on the combined 30 million images, as described in Section 4. We then cluster the selected SAE features in 100 topics (topic merging, see Section 4). After merging, we classify topics into two categories. Abstract topics are concerned with general image properties, such as mood, perspective, geometry, or layout. Concrete topics

| Num. Topics | 50 | | | 100 | | | 200 | | | 300 | | | 500 | | |
|---|---|---|---|---|---|---|---|---|---|---|---|---|---|---|---|
| | $C_I \uparrow$ | $C_R \uparrow$ | $D \uparrow$ | $C_I \uparrow$ | $C_R \uparrow$ | $D \uparrow$ | $C_I \uparrow$ | $C_R \uparrow$ | $D \uparrow$ | $C_I \uparrow$ | $C_R \uparrow$ | $D \uparrow$ | $C_I \uparrow$ | $C_R \uparrow$ | $D \uparrow$ |
| AVITM | 31.12 | 78.52 | 3.44 | 30.64 | 78.11 | 3.40 | 28.88 | 76.78 | 3.36 | 28.40 | 75.74 | 3.35 | 27.90 | 74.06 | 3.36 |
| CombinedTM | 42.30 | 79.39 | 3.82 | 35.11 | 59.65 | 3.82 | 23.16 | 30.80 | 3.80 | 21.62 | 28.74 | **3.83** | 20.29 | 26.56 | 3.80 |
| DecTM | 35.20 | 69.18 | **3.94** | 33.96 | 64.44 | **3.93** | 28.32 | 45.73 | **3.94** | 16.40 | 16.58 | 3.74 | 16.30 | 11.81 | 3.81 |
| DVAE | 16.93 | 5.75 | 3.64 | 16.06 | 5.64 | 3.51 | 16.94 | 10.05 | 3.20 | 16.08 | 6.20 | 3.16 | 16.25 | 7.44 | 3.11 |
| ETM | 20.46 | 42.15 | 3.43 | 19.44 | 35.63 | 3.51 | 19.33 | 28.85 | 3.58 | 18.67 | 24.45 | 3.62 | 18.25 | 19.93 | 3.67 |
| FASTopic | 34.44 | 69.56 | 3.54 | 33.00 | 70.06 | 3.54 | 32.28 | 68.14 | 3.57 | 32.68 | 69.94 | 3.55 | 31.05 | 67.27 | 3.53 |
| NSTM | 19.57 | 63.88 | 2.79 | 18.34 | 65.65 | 2.73 | 19.41 | 67.90 | 2.71 | 19.09 | 67.76 | 2.67 | 18.48 | 70.00 | 2.63 |
| TSCTM | 40.51 | 80.40 | 3.91 | 38.46 | 80.33 | 3.84 | 34.69 | 72.61 | **3.82** | 30.15 | 57.39 | 3.82 | 25.28 | 39.81 | **3.83** |
| SAE-TM (ours) | **42.57** | **85.05** | 3.70 | **40.11** | **85.67** | 3.54 | **38.59** | **85.53** | 3.54 | **37.57** | **85.05** | 3.53 | **36.54** | **84.43** | 3.53 |

*Table 3.* Performance on three image datasets. Values show topic coherence ($C_I$ = intruder detection accuracy, $C_R$ = topic coherence rating) and diversity ($D$) scores. All scores are averaged over datasets. Best values are in bold, and second-best values are underlined.

are concerned with objects visible in the images. In our analysis, we focus on concrete topics. For classification, we use an LLM (PHI-4; Abdin et al. (2024)). The same LLM also summarizes the top 20 words in each topic by emission probability. Additionally, we remove topics that activate for more than 30% of images (macro-average).

**Results.** In Figure 4, we show the top 10 topics sorted by the variance of their activity ratio across datasets. ImageNet has significantly more plants ("Delicate Plants") and animals ("Fluffy Animals", "Wildlife") than other datasets. ImageNet also has more technical tools ("Containers and Packaging"). In contrast, ImageNet has significantly fewer images showing humans ("Human Interaction"). These differences reflect the dataset construction. The class-balanced approach of ImageNet emphasizes animals and common objects, which are less frequent in the other three web-based datasets. However, differences also exist among them: CC3M and CC12M contain more text and typographic elements, a trend that is particularly pronounced for CC12M. YFCC features many urban scenes ("Urban Environment") and, with CC3M, many musical performances ("Live Performance"). Like ImageNet, YFCC also features many natural landscapes ("Lush Landscape"). These results show that dataset differences arise from construction, but even with similar methodology (CC3M, CC12M), differences emerge from varying image sources. Overall, we find that Sparse Autoencoders are an effective and efficient tool for understanding dataset composition, which directly motivates applications to trace downstream behavior, dataset rebalancing, and data selection. Importantly, SAE topic models avoid the need for expensive attribute labeling via MLLMs or specialized models (Zhang et al., 2024; Huang et al., 2023). Our analysis can be easily expanded by considering more or finer-grained topics, down to atomic SAE features.

### 6.2. Analyzing Topic Distribution in Artworks

We analyze how prevalent topics evolve in Japanese woodblock prints (177,897 images) provided by Khan & van

Noord (2021). This demonstrates applications of topic modeling in the humanities, specifically how deep learning research can enhance our understanding of cultural assets. We create captions for all images using InternVL3.5-14B and apply our pretrained text SAE to GRANITE-R2 embeddings of the captions, as we find that complex themes in art are poorly represented by SigLIP embeddings.

**Results.** As in Section 6.1, we show the 10 topics with the highest variance across categories. In Figure 5, we present the resulting topic proportions for four of the seven eras (images are categorized into seven eras spanning from the 1740s to the present) to illustrate trends (results for eras earlier than 1800 and later than 1950 are not shown to enhance clarity, but follow the observed trends).

Topics can be subdivided into four groups: Domestic scenes ("Domestic Scene", "Elderly Woman", and "Group Portrait"), Nature ("Rural Landscape", "Pine Trees", "Water Bodies", and "Coastal Scene"), Building ("Architectural Structure"), and Fashion/Accessories ("Vibrant Garment" and "Kimono Design"). Depictions of domestic scenes are particularly prevalent in the "Golden Age of Ukiyo-e (1780 to 1804)", and continually declined in popularity afterwards. Regarding fashion design, a clear divide is evident between the 20th century and the Edo and Meiji periods. Traditional attire is depicted much less frequently in the 20th century than before, reflecting changing customs and increasing Western influence. Finally, depictions of natural scenes and architecture experienced a significant increase in popularity in 20th-century woodblock prints. However, they were also relatively more popular at the end of the Edo period (1804-1868) compared to earlier periods and remained popular during the Meiji era. This reflects a shift from woodblock prints that focused on human relations and social scenes to those that featured nature and landscapes.

This analysis already reveals interesting trends that can be further expanded by considering more fine-grained topics, which is easily achieved given the modular design of SAE-

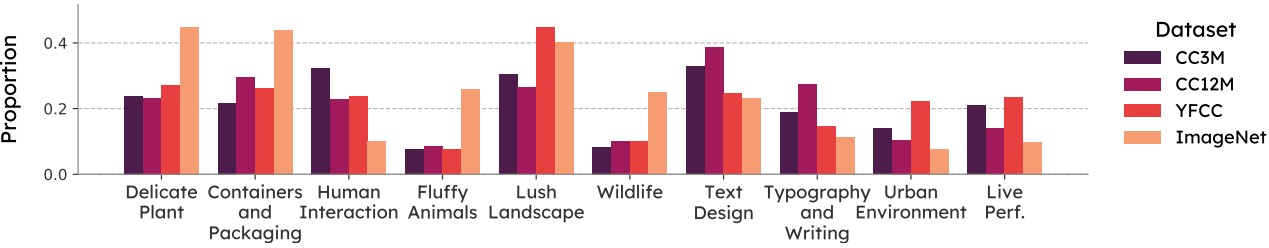

*Figure 4.* Statistics of top 10 topics with the highest variance across four popular image datasets. Values indicate the proportion of images in each dataset where the topic is active (even weakly). Differences between datasets reveal interesting trends, such as a comparatively higher frequency of images of animals and plants in ImageNet compared to web-sourced datasets.

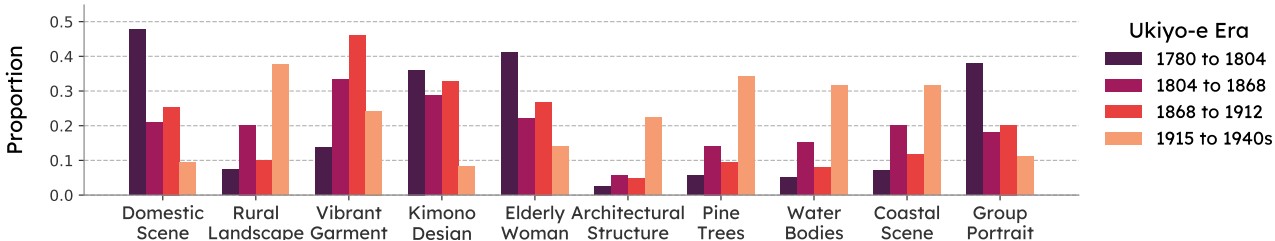

*Figure 5.* Statistics of top 10 topics with the highest variance in Japanese woodblock prints from different artistic periods. Changes in topic distribution reflect changing cultural environment (e.g., clothing) and popular themes (e.g., domestic scenes vs. nature).

derived topics. In summary, topic modeling is a useful tool for understanding complex image datasets, such as art, and our proposed SAE topic model is a well-suited method.

## 7. Conclusion

We have shown that SAEs can be understood as topic models by introducing a continuous topic model inspired by LDA and deriving the SAE objective as the corresponding MAP estimator under specific conditions. Under this view, SAE features function as thematic components that combine to explain embeddings, rather than as monosemantic or steerable mechanisms. Building on this, we proposed SAE-TMs, a framework in which SAEs are pretrained once to learn reusable topic atoms and later interpreted and merged for downstream datasets. Our experiments show that SAE-TMs yield coherent and diverse topics across five text and three image datasets, outperforming strong Neural Topic Modeling baselines. We further use SAE-TMs to compare thematic structure across image datasets and to analyze changes in themes in woodblock prints.

Concretely, SAE-TMs make five contributions to topic modeling. First, ours is the first derivation that casts the SAE objective as MAP inference under a topic model, providing a principled grounding for SAEs as topic-discovery tools. Second, the topic count $K'$ can be varied freely after pretraining via merging, with no retraining required. Third, SAE-TMs operate entirely in latent embedding space. They reconstruct the embedding rather than a bag-of-words, making them easily applicable across modalities. Fourth, SAE-TMs de-

couple topic learning (the frozen SAE) from interpretation (post-hoc word-emission learning), giving practitioners flexibility in how topics are represented. Fifth, every merged topic is transparently composed of its fine-grained atoms, which supports interpretability at multiple scales.

However, some limitations remain. SAE feature interpretation can be improved, and activation strengths do not always align with topic importance. Document embeddings also encode non-thematic information, such as sentiment, document length, or stylistic features, which SAE-TMs may encode alongside thematic structure. However, this is a property shared by all embedding-based topic models, including CombinedTM and FASTopic, and is not specific to our approach.

Several extensions of the SAE-TM framework follow naturally from our analysis. Assumption (A3) of Section 3.3 restricts our derivation to SAE variants with independent features, but recent hierarchical SAEs, such as Matryoshka SAEs (Nabeshima, 2024) that train nested codebooks at multiple scales, and Matching-Pursuit SAEs (Costa et al., 2025) that explicitly model hierarchical structure over directions, relax this assumption and suggest extensions toward hierarchical topic models. We see formalizing this connection and developing the corresponding inference procedure as a promising direction for future work.

Taken together, this work presents a unified theoretical framework, a practical modeling approach, and applications that underscore an effective role for SAEs in data analysis and representation research.

## Impact Statement

This paper presents foundational work on theoretically understanding Sparse Autoencoders with practical applications aimed at understanding the thematic structure of large-scale datasets. By formalizing the connection between SAEs and Topic Models, our research contributes to better insights into the properties of widely used interpretability tools and to the development of scalable methods to audit training data for potential biases and distributional differences. Furthermore, we demonstrate the utility of these methods in the digital humanities, supporting the analysis of cultural heritage materials.

However, there are ethical considerations regarding the reliability of automated interpretations. Our method relies on associating features with word distributions from existing textual descriptions or LLM-generated captions to assign meaning to topics. Biases or hallucinations present in these documents (e.g., stemming from captioning models but also biases of human annotators) could propagate into the resulting topic models, potentially leading to misrepresentations of data groups. Additionally, while our approach helps in discovering unknown themes, activation strengths do not always perfectly align with semantic importance. Therefore, users should exercise caution and perform qualitative validation when using these tools to draw conclusions about sensitive or sociological datasets.

### Acknowledgements

This work was partially funded by the ERC (853489 - DEXIM) and the Alfried Krupp von Bohlen und Halbach Foundation, for which we thank them for their generous support. The authors gratefully acknowledge the scientific support and resources of the AI service infrastructure *LRZ AI Systems* provided by the Leibniz Supercomputing Centre (LRZ) of the Bavarian Academy of Sciences and Humanities (BAdW), funded by Bayerisches Staatsministerium für Wissenschaft und Kunst (StMWK). We also acknowledge the use of the HPC cluster at Helmholtz Munich for the computational resources used in this study.

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

# Supplementary Material

## A. Relating Fixed-Sparsity SAEs to the CTM

Here, we relate fixed-sparsity SAEs, such as TopK (Gao et al., 2025) and BatchTopK (Bussmann et al., 2024), to the continuous topic model (CTM) introduced in Section 3.2. In contrast to the $L_1$-penalty formulation analyzed in Section 3.3, fixed-sparsity SAEs enforce a hard limit on the number of active features. We show that they arise from a deterministic support-selection approximation to MAP inference under the CTM.

### Aggregated topic activations

Let the topic directions form the decoder matrix ($W = [\mu_1, \ldots, \mu_K] \in \mathbb{R}^{d \times K}$). In the CTM, an embedding is generated by contributions ($c_n = \lambda_n w_n$), where ($w_n \sim \mathcal{N}(\mu_{z_n}, \Sigma_{z_n})$). Grouping contributions by topic yields aggregated activations

$$a_k := \sum_{n:z_n=k} \lambda_n, \qquad a = (a_1, \ldots, a_K)^\top \in \mathbb{R}^K_{\geq 0}. \tag{14}$$

Under concentrated directions ($\Sigma_k \to 0$), $w_n \approx \mu_{z_n}$, and hence

$$D \mid a \sim \mathcal{N}(Wa, \sigma^2 I), \tag{15}$$

recovering the standard SAE reconstruction model.

### Prior over activations

For topic $k$, let $N_k \sim \text{Pois}(\rho_d \theta_k)$ denote the number of contributions, and let each contribution strength follow $\lambda_{k,i} \sim \text{Ga}(r_k, \tau_k)$. Then

$$a_k = \sum_{i=1}^{N_k} \lambda_{k,i} \tag{16}$$

is compound-Poisson-Gamma distributed, featuring a point mass at zero (topic inactive) and a continuous density for $a_k > 0$. For the special case $r_k = 1$, this density takes the closed form

$$f_{a_k}(a) = e^{-(\rho_k + \tau_k a)} \sqrt{\frac{\rho_k \tau_k}{a}} I_1\left(2\sqrt{\rho_k \tau_k a}\right), \qquad a > 0, \tag{17}$$

where $\rho_k = \rho_d \theta_k$ and $I_1(\cdot)$ is the modified Bessel function of the first kind. The MAP objective for a single embedding is

$$\mathcal{L}(a) = \frac{1}{2\sigma^2} \|D - Wa\|_2^2 + \sum_{k=1}^{K} \left[-\log p(a_k)\right]. \tag{18}$$

For $a_k = 0$, the penalty equals $\rho_k$. For $a_k > 0$, the dominant term $\tau_k a_k$ induces magnitude shrinkage.

### Deterministic support as Inference Approximation

The exact MAP objective in Equation (18) is computationally intractable, as it requires a combinatorial search over the sparse support (the "spike" vs. "slab") of the compound-Poisson-Gamma prior. Fixed-sparsity SAEs therefore use the encoder to find an *approximate* MAP solution via deterministic support selection. Instead of sampling $N_k$ or penalizing $|a_k|$, the encoder chooses a subset of indices $\mathcal{S} \subset \{1, \ldots, K\}$ of fixed size $|\mathcal{S}| = k \ll K$, sets $a_j = 0$ for $j \notin \mathcal{S}$, and computes the remaining $a_j$ directly via the encoder. This corresponds to:

$$-\log p(a) \approx \text{const} \quad \text{subject to } |\{k : a_k > 0\}| = k, \tag{19}$$

This approximation effectively replaces the complex CPG prior with a constant $L_0$ penalty. This penalty is simply the fixed hyperparameter $k$ (the number of active features), which is not learned via the objective but set externally, mirroring standard SAE training. This is, in other words, a hard constraint on the number of active topics rather than a soft prior over magnitudes.

**Summary**

Fixed-sparsity SAEs arise from the CTM under the following simplifications:

- *Concentrated topic directions:* $\Sigma_k \to 0$, yielding the linear decoder $W = [\mu_1, \ldots, \mu_K]$.
- *Deterministic support selection:* Approximate the intractable MAP inference under the CPG prior by directly choosing a fixed active set.
- *Low effective activity:* Use $k \ll K$, mirroring the small-$\rho_d$ regime where only a few topics contribute.

Thus, fixed-sparsity SAEs correspond to MAP inference in the continuous topic model with a deterministic sparse-support constraint. This connection clarifies how decoder weights and sparsity levels correspond to the generative quantities in Section 3.2, and explains why fixed-sparsity SAEs behave as topic models in practice.

# B. SAE Pretraining

**Text SAE.** We pretrain a foundational SAE for text on a large dataset combining sections from Wikipedia and C4 (Raffel et al., 2020). Concretely, we sample 240 million sections each from Wikipedia and C4 and embed them using `granite-r2` (Awasthy et al., 2025). A section is a sequence of $n \in [1, \ldots, 10]$ consecutive sentences within a single document. In Wikipedia, we treat individual paragraphs in articles as documents and only consider paragraphs with at least 5 sentences. In C4, we consider all documents. Using the combined 480 million embedded sections, we train a BatchTopK SAE (Bussmann et al., 2024) with an expansion factor of 64 (dictionary size is $49,152$ features) and 32 active features per embedding for 800,000 steps with a batch size of 4,096. The SAE achieves reconstruction $R^2 = 0.785$.

**Image SAE.** We pretrain a SAE for images on SigLIP embeddings of 360 million images in LAION-400M (Schuhmann et al., 2021). Concretely, we train a BatchTopK SAE for 50,000 steps with batch size 20,000 on image embeddings of VIT-B-16-SIGLIP from OpenCLIP (Ilharco et al., 2021). The SAE has expansion factor of 32 and 32 active features per embedding. It achieved reconstruction $R^2 = 0.872$.

# C. Detailed Description of the Topic Merging Procedure

The goal of this procedure is merge the large number $K$ of sparse, fine-grained SAE features ("topic atoms") into a smaller set of $K' \ll K$ distinct topics. The process uses word embeddings to cluster features that activate on semantically similar words, weighted by the frequency of feature activation. Pseudocode for the algorithm is in Algorithm 1.

### C.1. Feature Representation (Topic Atom Embeddings)

Each SAE feature $k$ is characterized by its word emission probabilities $B_k \in \mathbb{R}^V$, representing the probability distribution $P(w|k)$ over the vocabulary. To compute a dense vector representation $\mathbf{T}_k$ for feature $k$:

1. **Denoising (Top-$p$ Filtering):** We filter the distribution $B_k$ to retain only the most significant words. We select the smallest set of words whose cumulative probability mass sums to at least $p$ (in our experiments, we choose $p = 0.9$).

2. **Renormalization:** The surviving entries are renormalized to sum to 1, forming a sparse vector $\hat{B}_k$.

3. **Projection:** We compute the weighted sum of word embeddings: $\mathbf{T}_k = \sum_{w_i \in \mathcal{V}} \hat{B}_{k,i} \cdot \mathbf{w}_i$, where $\mathbf{w}_i$ is the embedding vector for word $w_i$.

### C.2. Weighted Clustering

We cluster the topic atom embeddings $\{\mathbf{T}_k\}_{k=1}^K$ into $K'$ groups.

- **Algorithm:** We employ $k$-means clustering on the feature embeddings.

- **Weighting:** The clustering is weighted by the global importance of each feature, given by $P(k)$ (the average activation value of feature $k$ across the dataset). This ensures that frequent features drive the formation of cluster centroids more than rare features.

## C.3. Topic Aggregation

Once cluster assignments $c_k \in \{1, \ldots, K'\}$ are obtained, we aggregate the features in each cluster $k'$ to form the final topic distribution $P_{k'}(w)$. This is computed as the weighted average of the original word emission distributions $B_k$, weighted by their activation frequency $P(k)$, as shown in Equation (1) of the main text.

---

**Algorithm 1:** SAE Topic Merging

**Input:** Word Emission Matrix $B \in \mathbb{R}^{K \times V}$ (SAE Features $\times$ Vocab), Feature Frequencies $\mathbf{p}_{feat} \in \mathbb{R}^K$ (where $\mathbf{p}_{feat}[k] = P(k)$), Word Embeddings $\mathbf{W} \in \mathbb{R}^{V \times D}$, Doc-Feature Matrix $\Theta_{csr}$, Cumulative Probability Threshold $p$ (e.g., 0.9), Target Topic Count $K'$

**Output:** Cluster Assignments $\mathcal{C}$, Aggregated Topic Distributions $\mathbf{B}'$

1   **Function** `SparsifyAndRenormalize`$(M, p)$**:**

     // Keeps minimal top words summing to probability mass p

2     Initialize $M_{out} \leftarrow \mathbf{0}^{K \times V}$

3     **for** *each feature $k$ in $M$* **do**

4        Sort indices $idx$ such that $M[k, idx]$ is descending

5        Calculate cumulative sum $S = \text{cumsum}(M[k, idx])$

6        Find cut-off index $n = \min\{i \mid S[i] > p\}$

          // Keep top n elements, mask others

7        Set active indices $A \leftarrow idx[0 \ldots n]$

8        $M_{out}[k, A] \leftarrow M[k, A]$

          // Renormalize to sum to 1

9        $M_{out}[k, :] \leftarrow M_{out}[k, :] / \sum(M_{out}[k, :])$

10    **end**

11    **return** $M_{out}$

12   **Procedure** `Main`**:**

     // 1.  Filter Dead Features

13    Find active indices $\mathcal{K}_{valid} = \{k \mid \mathbf{p}_{feat}[k] > 0\}$

14    $B_{valid} \leftarrow B[\mathcal{K}_{valid}, :], P_{valid} \leftarrow \mathbf{p}_{feat}[\mathcal{K}_{valid}]$

     // 2.  Generate Topic Atom Embeddings

15    $B_{sparse} \leftarrow \text{SparsifyAndRenormalize}(B_{valid}, p)$

     /* Calculate weighted average of word embeddings for each SAE feature   */

16    $\mathbf{T} \leftarrow B_{sparse} \times \mathbf{W}$                       // $\mathbf{T}_k = \sum B_{k,i} \cdot \mathbf{w}_i$

     // 3.  Weighted Clustering

17    Initialize KMeans with $K'$ clusters

     /* Fit embeddings $\mathbf{T}$ weighted by feature frequency $P(k)$                */

18    $\mathcal{L} \leftarrow \text{KMeans}(X = \mathbf{T}, \text{weights} = P_{valid})$

     // 4.  Merge Topics (Aggregation)

19    Initialize $\mathcal{C}$ mapping Cluster ID $k' \rightarrow$ Feature Indices $k$

20    Initialize $\mathbf{B}' \in \mathbb{R}^{K' \times V}$

21    **for** $k' \leftarrow 0$ **to** $K' - 1$ **do**

22        Get features in cluster: $\mathcal{F}_{k'} = \{k \in \mathcal{K}_{valid} \mid \mathcal{L}[k] = k'\}$

          /* Compute aggregated distribution: Eq. (1)                         */

23        $Numerator \leftarrow \sum_{k \in \mathcal{F}_{k'}} (B_{valid}[k] \cdot P_{valid}[k])$

24        $Denominator \leftarrow \sum_{k \in \mathcal{F}_{k'}} P_{valid}[k]$

25        $\mathbf{B}'[k'] \leftarrow Numerator / Denominator$

26    **end**

27    **return** $\mathcal{C}, \mathbf{B}'$

---

## D. Topic Diversity: Ratio of Unique Words

In this section, we show the topic diversity for text-only datasets, corresponding to Table 2, but use the ratio of unique words (Dieng et al., 2020) as the metric. The results are in Table 4. We observe that the overall trends are similar, but the ratio of unique words decreases as the number of topics increases. This occurs because the total number of words considered for topic evaluation scales with $n \times k$, where $n = 20$ in our case (see Section 5). Since we use only the top 5000 most common lemmas in each dataset for topic model training, the number of unique words is capped. Therefore, the decreasing trend in the metric is mathematically inevitable. In contrast, the WMD metric used in the main paper does not suffer from this issue and permits comparison across different numbers of topics.

|  | Number of Topics | | | | |
|---|---|---|---|---|---|
|  | 50 | 100 | 200 | 300 | 500 |
| AVITM | 0.623 | 0.443 | 0.260 | 0.209 | 0.135 |
| CombinedTM | 0.637 | 0.486 | 0.385 | 0.362 | 0.233 |
| DecTM | 0.784 | 0.620 | 0.428 | 0.394 | 0.284 |
| DVAE | 0.485 | 0.319 | 0.199 | 0.141 | 0.096 |
| ETM | 0.769 | 0.646 | 0.509 | 0.426 | 0.328 |
| FASTopic | 0.639 | 0.586 | 0.527 | 0.451 | 0.326 |
| NSTM | 0.406 | 0.284 | 0.207 | 0.160 | 0.057 |
| TSCTM | **0.942** | **0.806** | **0.652** | **0.536** | **0.388** |
| SAE-TM (ours) | 0.731 | 0.592 | 0.447 | 0.369 | 0.279 |

*Table 4.* Topic diversity scores on text-only datasets (macro-averages of all 5 datasets), using the ratio of unique words (Dieng et al., 2020). Best values are in bold, and second-best values are underlined. Scores are not comparable for different numbers of topics, which is the case for the WMD metric in the main paper.

## E. Ablating the Background Prior Weight $\pi$

Here, we ablate the background prior weight $\pi$ introduced in Section 4 (Equation (11)) to learn more faithful SAE feature interpretations. We sweep values $\pi \in \{0.1, 0.2, \ldots, 0.9\}$ and evaluate the resulting interpretations using $C_R$ (topic relevance), $C_I$ (intruder detection), and $D$ (topic diversity) on all text datasets. The following table reports the results:

| $\pi$ | 50 | | | 100 | | | 200 | | | 300 | | | 500 | | |
|---|---|---|---|---|---|---|---|---|---|---|---|---|---|---|---|
|  | $C_I \uparrow$ | $C_R \uparrow$ | $D \uparrow$ | $C_I \uparrow$ | $C_R \uparrow$ | $D \uparrow$ | $C_I \uparrow$ | $C_R \uparrow$ | $D \uparrow$ | $C_I \uparrow$ | $C_R \uparrow$ | $D \uparrow$ | $C_I \uparrow$ | $C_R \uparrow$ | $D \uparrow$ |
| 0.1 | 52.69 | 77.75 | 3.58 | 48.44 | 77.05 | 3.56 | 45.45 | 76.67 | 3.52 | 43.19 | 75.05 | 3.50 | 39.29 | 72.19 | 3.47 |
| 0.2 | **56.30** | **78.16** | 3.65 | 51.31 | 77.70 | 3.62 | 47.13 | **76.92** | 3.59 | **45.23** | **75.86** | 3.57 | **41.79** | **73.13** | 3.55 |
| 0.3 | 54.53 | 77.30 | 3.67 | **52.01** | **78.25** | 3.64 | 46.16 | 75.94 | 3.60 | 43.75 | 74.28 | 3.58 | 40.56 | 71.43 | 3.57 |
| 0.4 | 55.30 | 77.53 | 3.70 | 50.71 | 77.03 | 3.67 | 46.77 | 75.20 | 3.64 | 43.88 | 73.41 | 3.62 | 40.95 | 71.01 | 3.60 |
| 0.5 | 54.13 | 75.30 | 3.71 | 51.54 | 76.43 | 3.71 | **47.34** | 74.49 | 3.68 | 44.46 | 72.44 | 3.66 | 41.24 | 70.07 | 3.64 |
| 0.6 | 53.26 | 74.57 | 3.73 | 50.49 | 75.87 | 3.73 | 47.11 | 73.47 | 3.71 | 44.86 | 72.05 | 3.70 | 41.21 | 68.81 | 3.68 |
| 0.7 | 54.38 | 75.49 | 3.78 | 50.72 | 74.65 | 3.76 | 46.63 | 72.02 | 3.75 | 43.76 | 70.19 | 3.74 | 40.55 | 67.23 | 3.72 |
| 0.8 | 53.81 | 72.57 | 3.80 | 49.38 | 71.22 | 3.81 | 45.59 | 68.97 | 3.79 | 42.99 | 67.03 | 3.78 | 39.79 | 64.65 | 3.77 |
| 0.9 | 52.98 | 69.88 | **3.86** | 49.92 | 69.50 | **3.86** | 43.85 | 66.03 | **3.85** | 40.91 | 63.06 | **3.85** | 38.05 | 60.29 | **3.83** |

*Table 5.* Ablation of the background-prior weight $\pi$ in Section 4, on text-only datasets (macro-averages across the five datasets of Section 5.1). Best values per column are in bold, second-best values are underlined.

We find that scores are stable across $\pi \in [0.2, 0.5]$, with coherence peaking near $\pi = 0.2$ and diversity increasing monotonically with $\pi$. The default $\pi = 0.3$ used in our main experiments lies within this stable range.

## F. Results on Topic Relevance

To demonstrate that the learned topics are not only coherent and diverse but also meaningful to the documents they represent, we evaluate topic relevance using an LLM (Phi-4). For a given document, we sample one active topic and one inactive

topic and prompt the LLM to decide whether each topic is relevant to the document. If a model's assigned topics are poorly aligned with the text, the LLM will not be able to distinguish between active and inactive topics.

Table 6 reports the overall topic relevance accuracy averaged across all text datasets. SAE-TM outperforms all baselines across all topic numbers, and achieves a mean accuracy of 66.5% compared to 61.7% for the best baseline, AVITM.

While baseline accuracy on inactive (irrelevant) topics is near 100%, as it is clear when topics are unrelated, active but relevant topics are often broad and thus harder to correctly attribute to the document. Table 7 shows the detection accuracy exclusively on active topics. Here, the performance margin is particularly favorable for SAE-TM, which correctly identifies active topics as relevant 38.3% of the time on average, substantially outperforming the strongest baseline (AVITM at 29.0%).

| Model | 50 | 100 | 200 | 300 | 500 | Mean |
|---|---|---|---|---|---|---|
| AVITM | 61.34 | 61.67 | 63.29 | 63.94 | 59.99 | 61.66 |
| CombinedTM | 58.05 | 58.61 | 58.19 | 53.37 | 55.56 | 56.73 |
| DecTM | 57.45 | 59.21 | 59.16 | 51.95 | 52.60 | 56.08 |
| DVAE | 49.84 | 49.88 | 49.68 | 49.79 | 49.90 | 49.74 |
| ETM | 52.89 | 53.55 | 52.83 | 52.65 | 52.61 | 52.88 |
| Fastopic | 55.25 | 54.62 | 55.44 | 57.14 | 57.65 | 56.26 |
| NSTM | 50.01 | 50.78 | 51.57 | 50.97 | 52.62 | 51.36 |
| TSCTM | 60.32 | 60.97 | 61.98 | 62.63 | 62.00 | 61.57 |
| SAE-TM | **64.14** | **63.74** | **66.49** | **67.54** | **69.36** | **66.51** |

*Table 6.* Overall topic relevance accuracy (mean across text datasets). Best results are bolded.

| Model | 50 | 100 | 200 | 300 | 500 | Mean |
|---|---|---|---|---|---|---|
| AVITM | 27.76 | 27.87 | 31.77 | 32.66 | 25.05 | 29.00 |
| CombinedTM | 17.30 | 18.45 | 17.18 | 7.75 | 12.45 | 14.68 |
| DecTM | 16.10 | 20.39 | 21.34 | 6.41 | 6.41 | 13.99 |
| DVAE | 0.33 | 1.54 | 0.37 | 0.24 | 0.75 | 0.58 |
| ETM | 8.18 | 9.27 | 7.65 | 6.95 | 6.89 | 7.66 |
| Fastopic | 13.94 | 12.03 | 13.30 | 16.14 | 17.71 | 15.16 |
| NSTM | 6.94 | 5.58 | 7.46 | 6.37 | 7.96 | 6.93 |
| TSCTM | 21.42 | 22.90 | 25.16 | 26.44 | 24.88 | 24.15 |
| SAE-TM | **33.65** | **33.09** | **37.95** | **40.32** | **43.69** | **38.26** |

*Table 7.* Overall detection accuracy on active topics only (mean across text datasets). Best results are bolded.

## G. Prompts

> **Image Captioning Prompt**
>
> You are an expert image analyst and descriptive writer specializing in creating "dense captions." Your task is to generate a single, continuous paragraph of highly detailed and comprehensive text that describes the provided image. Your description must be objective and based solely on visual evidence.
>
> Follow this multi-step process for your analysis:
>
> 1. Holistic Overview: Begin by establishing the overall scene. Describe the setting (e.g., urban street, natural landscape, indoor room), the time of day (e.g., midday, golden hour, night), the overall atmosphere or mood (e.g., bustling, serene, melancholic), and the general color palette.
>
> 2. Primary Subjects and Actions: Identify and describe the primary subject(s) in detail. If they are people, describe their apparent age, gender, clothing, posture, expression, and any actions they are performing. If they are objects or animals, describe their type, condition, color, and position. Describe the interactions between primary subjects.

3. Secondary Elements and Background: Detail the secondary subjects, significant objects, and the immediate background. Describe architectural elements, furniture, vehicles, flora, and fauna that populate the scene but are not the central focus. Describe their spatial relationship to the primary subjects.

4. Fine-Grained Details and Textures: Scrutinize the image for fine-grained details. Mention specific textures (e.g., the rough bark of a tree, the smooth surface of a metal table, the fabric weave of a coat), small, easily missed objects, text or symbols visible on signs or clothing, reflections in windows or water, and the quality of light and shadow (e.g., sharp, defined shadows indicating harsh light, or soft, diffuse light).

5. Synthesis and Composition: Conclude by synthesizing all observations. Briefly describe the photographic composition, such as the framing, perspective, and depth of field (e.g., a shallow depth of field blurring the background, a wide-angle shot capturing a vast landscape).

Formatting and Style Constraints:

- Output Format: Your entire output must be one single, continuous paragraph.
- No Line Breaks: Do not use any line breaks, newlines, or paragraph breaks (\n).
- Style: Write in a descriptive, objective, and formal tone.
- Exclusions:
    - Do not start with phrases like "This is an image of," "The picture shows," or any similar introductory statement.
    - Do not include personal opinions, judgments, or interpretations that are not directly supported by visual evidence.
    - Do not use bullet points, lists, or headers in your final output. Your entire response must be the caption itself.

Example of Desired Output:

A vibrant and crowded marketplace unfolds under the bright, hazy sun of midday, characterized by a dominant palette of warm ochres, deep reds, and earthen browns. The central focus is a male vendor in his late fifties, wearing a light blue djellaba and a straw hat, who is carefully arranging a pyramid of colorful spices on a rough wooden stall; his face is weathered and creased in concentration. In front of him, a tourist with a backpack slung over one shoulder, clad in a khaki shirt, points at a specific spice mound while a young child clings to her hand, looking with wide eyes at a nearby stall selling intricately woven leather bags. The background is a dense tapestry of activity, with other shoppers and vendors creating a soft-focus blur of movement, set against the backdrop of ancient, reddish-pink plaster walls and arched doorways. Fine details abound, from the coarse texture of the burlap sacks holding grains and the gleam of polished brass lanterns hanging from a wooden beam, to the subtle shadows cast by the woven canopy overhead, dappling the ground in a shifting pattern of light. The composition is tight and layered, creating a deep sense of immersion and chaotic energy, capturing the scene from a slightly low, eye-level perspective that places the viewer directly within the bustling alleyway.

Your Task:

Now, analyze the following image and generate the dense caption, strictly adhering to all instructions above.

---

### Image Captioning Prompt (for Ukiyo-e Images)

You are given an image of a Japanese woodblock print. Provide a single continuous paragraph of detailed analysis that follows these instructions: Describe the visual scene in objective, scientific, and precise language. Identify and name all visible figures, landmarks, buildings, or natural features if they are recognizable and culturally significant, and state their role in the composition. Describe the arrangement and interaction of figures, objects, and background elements. Note any inscriptions, seals, or cartouches, including their placement, without attempting speculative translation. Mention stylistic or technical aspects that are clearly visible, such as color layering, printing techniques, or use of pattern. Interpret the cultural or historical significance of the depicted subject only when directly inferable from visible attributes, without speculation or reference to things outside the image. If an element's identity or meaning is uncertain, explicitly state that it cannot be determined. Do not use subjective adjectives like "beautiful" or "ethereal", and do not mention arbitrary concepts not present in the image. Ensure that the analysis is precise, objective, culturally informed, and presented as one continuous paragraph without lists, headings, or bullet points.

**LLM-as-a-judge for Intruder Detection**

From the following list of words, identify the single word that does not belong with the others. The words are: {words}.

Your response must be only the single intruder word and nothing else.

**LLM-as-a-judge for Coherence Rating**

You are an expert in semantics and lexical relationships. Your task is to evaluate the coherence of the following list of words: '{words}'.

Coherence is how well the words belong to a single, clear, and specific category.

- A score of 100 means the words are extremely coherent (e.g., all are types of citrus fruits).
- A score around 50 means the words are moderately coherent (e.g., all are 'vehicles' but mix cars, boats, and planes).
- A score of 0 means the words are completely unrelated.

Provide your analysis as a JSON object with two keys: "rationale" and "score".

- "rationale": A brief, one-sentence explanation for your score.
- "score": An integer between 0 and 100.

Your response MUST be only the JSON object and nothing else."

# H. Full Dataset Results

We report the full per-dataset results, i.e. an expanded version of Table 2, for text datasets in Table 8. Additionally, we report the full per-dataset results (expanded version of Table 3) for image dataset in Table 9.

| Num. Topics | 50 | | | 100 | | | 200 | | | 300 | | | 500 | | |
|---|---|---|---|---|---|---|---|---|---|---|---|---|---|---|---|
| | $C_I \uparrow$ | $C_R \uparrow$ | $D \uparrow$ | $C_I \uparrow$ | $C_R \uparrow$ | $D \uparrow$ | $C_I \uparrow$ | $C_R \uparrow$ | $D \uparrow$ | $C_I \uparrow$ | $C_R \uparrow$ | $D \uparrow$ | $C_I \uparrow$ | $C_R \uparrow$ | $D \uparrow$ |
| **News-20k** | | | | | | | | | | | | | | | |
| AVITM (Srivastava & Sutton, 2017) | 39.76 | 66.43 | 3.47 | 36.08 | 66.15 | 3.42 | 36.60 | 64.63 | 3.50 | 34.57 | 60.03 | 3.61 | 33.67 | 60.57 | 3.47 |
| CombinedTM (Bianchi et al., 2021a) | 45.64 | 71.46 | **3.97** | 42.26 | 66.74 | **3.93** | **44.05** | **70.02** | **3.96** | **48.09** | **73.51** | **3.95** | **43.46** | **68.54** | **3.93** |
| DecTM (Wu et al., 2021) | 39.68 | 60.32 | 3.90 | 35.86 | 59.88 | 3.87 | 33.95 | 57.93 | 3.81 | 32.04 | 56.36 | 3.81 | 30.12 | 53.56 | 3.78 |
| DVAE (Burkhardt & Kramer, 2019) | 22.32 | 27.02 | 3.66 | 18.54 | 5.34 | 3.37 | 16.71 | 3.18 | 3.37 | 17.49 | 7.05 | 3.41 | 16.56 | 2.00 | 3.38 |
| ETM (Dieng et al., 2020) | 21.64 | 17.79 | 3.16 | 18.48 | 14.65 | 3.32 | 19.47 | 10.91 | 3.44 | 18.87 | 11.79 | 3.47 | 17.88 | 8.81 | 3.52 |
| FASTopic (Wu et al., 2024b) | 34.04 | 59.76 | 3.36 | 31.66 | 56.16 | 3.48 | 31.24 | 58.76 | 3.44 | 29.52 | 55.23 | 3.43 | 28.78 | 58.29 | 3.37 |
| NSTM (Zhao et al., 2021) | 26.04 | 42.90 | 3.22 | 27.18 | 52.10 | 3.16 | 28.71 | 56.19 | 3.11 | 29.61 | 55.90 | 3.13 | 29.80 | 54.76 | 3.12 |
| TSCTM (Wu et al., 2022) | 37.16 | 60.02 | 3.81 | 33.38 | 55.62 | 3.81 | 26.86 | 34.60 | 3.77 | 23.59 | 23.43 | 3.72 | 20.58 | 14.80 | 3.68 |
| SAE-TM (ours) | **49.16** | **71.91** | 3.61 | **48.06** | **75.33** | 3.62 | 41.32 | **72.93** | 3.58 | 39.25 | 72.35 | 3.57 | 36.76 | **68.74** | 3.53 |
| **IMDB** | | | | | | | | | | | | | | | |
| AVITM (Srivastava & Sutton, 2017) | 31.84 | 67.14 | 3.07 | 30.44 | 68.64 | 3.07 | 28.37 | 66.23 | 3.05 | 28.05 | 65.60 | 3.04 | 22.73 | 49.27 | 2.64 |
| CombinedTM (Bianchi et al., 2021a) | 37.88 | 66.54 | **3.74** | 33.64 | 63.87 | 3.69 | 32.10 | 60.57 | 3.66 | 16.56 | 4.05 | **3.65** | 16.88 | 4.66 | 3.64 |
| DecTM (Wu et al., 2021) | 37.64 | 63.09 | 3.71 | 34.98 | 61.46 | **3.77** | 19.42 | 12.41 | 3.61 | 20.65 | 18.41 | 3.57 | 16.82 | 6.35 | 3.64 |
| DVAE (Burkhardt & Kramer, 2019) | 16.20 | 4.56 | 3.26 | 16.72 | 2.63 | 3.30 | 17.50 | 2.66 | 3.08 | 15.38 | 5.80 | 3.11 | 16.45 | 3.57 | 3.05 |
| ETM (Dieng et al., 2020) | 20.20 | 30.65 | 3.02 | 20.00 | 23.10 | 3.13 | 19.62 | 17.34 | 3.19 | 19.45 | 16.56 | 3.24 | 19.03 | 12.85 | 3.32 |
| FASTopic (Wu et al., 2024b) | 31.12 | 54.42 | 1.74 | 27.66 | 46.51 | 1.94 | 26.21 | 39.91 | 2.28 | 26.08 | 51.15 | 2.64 | 25.54 | 46.45 | 2.80 |
| NSTM (Zhao et al., 2021) | 16.76 | 39.99 | 2.79 | 20.10 | 41.95 | 2.73 | 18.19 | 47.99 | 2.61 | 18.50 | 46.49 | 2.60 | 17.26 | 43.40 | 2.49 |
| TSCTM (Wu et al., 2022) | 35.64 | 65.22 | 3.73 | 27.44 | 48.23 | 3.68 | 23.30 | 33.55 | 3.69 | 20.63 | 17.34 | 3.64 | 18.92 | 14.15 | 3.64 |
| SAE-TM (ours) | **51.04** | **79.04** | 3.65 | **45.22** | **74.87** | 3.57 | **41.29** | **72.47** | 3.52 | **38.32** | **70.56** | 3.50 | **34.96** | **66.95** | 3.49 |
| **Yelp** | | | | | | | | | | | | | | | |
| AVITM (Srivastava & Sutton, 2017) | 37.00 | 69.00 | 3.25 | 35.06 | 74.59 | 3.20 | 33.14 | 71.89 | 3.20 | 32.69 | 70.28 | 3.17 | 31.60 | 68.48 | 3.15 |
| CombinedTM (Bianchi et al., 2021a) | **49.36** | 72.71 | 4.02 | 49.68 | 69.79 | **4.00** | 48.76 | 68.31 | **3.99** | 16.45 | 3.56 | 3.75 | **43.25** | 66.44 | **3.89** |
| DecTM (Wu et al., 2021) | 48.68 | 70.80 | **4.05** | 41.24 | 65.96 | 3.94 | 29.09 | 49.99 | 3.90 | 17.38 | 2.45 | 3.72 | 17.32 | 1.97 | 3.77 |
| DVAE (Burkhardt & Kramer, 2019) | 18.96 | 1.79 | 3.63 | 16.92 | 3.59 | 3.21 | 16.49 | 1.32 | 3.15 | 17.19 | 1.53 | 3.10 | 16.99 | 0.98 | 2.97 |
| ETM (Dieng et al., 2020) | 19.80 | 29.69 | 3.12 | 19.92 | 20.50 | 3.21 | 19.32 | 16.08 | 3.32 | 18.37 | 13.46 | 3.38 | 17.84 | 10.40 | 3.44 |
| FASTopic (Wu et al., 2024b) | 30.44 | 45.88 | 3.54 | 30.06 | 42.09 | 3.44 | 30.69 | 44.41 | 3.44 | 28.48 | 38.82 | 3.30 | 27.60 | 36.58 | 3.23 |
| NSTM (Zhao et al., 2021) | 21.96 | 50.99 | 3.03 | 24.94 | 55.63 | 2.95 | 24.53 | 51.28 | 2.92 | 25.11 | 54.12 | 2.97 | 24.67 | 49.17 | 3.06 |
| TSCTM (Wu et al., 2022) | 48.56 | 74.13 | 3.97 | 33.48 | 51.91 | 3.85 | 26.85 | 32.53 | 3.80 | 23.23 | 21.88 | 3.76 | 20.98 | 14.32 | 3.74 |
| SAE-TM (ours) | 48.80 | **81.11** | 3.58 | 47.60 | **80.68** | 3.55 | 43.87 | **77.33** | 3.52 | **40.56** | **75.86** | 3.51 | 37.93 | **72.19** | 3.51 |
| **Dailymail** | | | | | | | | | | | | | | | |
| AVITM (Srivastava & Sutton, 2017) | 47.24 | 82.67 | 3.48 | 41.68 | 82.31 | 3.49 | 38.31 | 78.73 | 3.47 | 36.83 | 75.87 | 3.49 | 33.35 | 72.88 | 3.45 |
| CombinedTM (Bianchi et al., 2021a) | 54.80 | 79.07 | **4.09** | 50.14 | 77.83 | **4.03** | 44.51 | 74.25 | **3.99** | 40.27 | 71.69 | **3.99** | 38.28 | 69.46 | **3.99** |
| DecTM (Wu et al., 2021) | 51.64 | 77.12 | 4.07 | 48.58 | 81.32 | 3.97 | 45.70 | 84.08 | 3.86 | 38.71 | 72.61 | 3.81 | 36.07 | 67.97 | 3.76 |
| DVAE (Burkhardt & Kramer, 2019) | 31.60 | 77.88 | 0.69 | 18.34 | 36.02 | 3.20 | 16.73 | 1.61 | 3.41 | 16.59 | 2.45 | 3.28 | 15.70 | 1.13 | 3.37 |
| ETM (Dieng et al., 2020) | 23.28 | 35.52 | 3.16 | 20.38 | 30.42 | 3.24 | 20.08 | 31.40 | 3.27 | 20.61 | 29.61 | 3.26 | 20.60 | 27.32 | 3.24 |
| FASTopic (Wu et al., 2024b) | 33.64 | 64.16 | 3.02 | 31.94 | 68.03 | 2.99 | 31.79 | 64.24 | 2.42 | 28.87 | 64.14 | 2.50 | 30.32 | 63.69 | 2.47 |
| NSTM (Zhao et al., 2021) | 26.12 | 57.57 | 2.74 | 24.66 | 58.75 | 2.68 | 24.16 | 50.88 | 2.59 | 23.90 | 54.36 | 2.55 | 22.00 | 47.59 | 2.37 |
| TSCTM (Wu et al., 2022) | 57.32 | **86.09** | 4.05 | 45.60 | 73.35 | 3.98 | 35.15 | 49.24 | 3.88 | 30.51 | 36.22 | 3.81 | 25.26 | 24.48 | 3.77 |
| SAE-TM (ours) | **67.16** | 83.64 | 3.82 | **62.06** | 83.58 | 3.79 | **60.78** | 85.43 | 3.74 | **57.51** | 85.63 | 3.70 | **54.15** | 83.58 | 3.67 |
| **Twitter** | | | | | | | | | | | | | | | |
| AVITM (Srivastava & Sutton, 2017) | 37.76 | 60.00 | 3.53 | 35.42 | 58.95 | 3.60 | 36.54 | 59.38 | 3.15 | 34.76 | 56.55 | 3.06 | 34.05 | 55.71 | 2.92 |
| CombinedTM (Bianchi et al., 2021a) | 16.80 | 61.41 | 0.40 | 16.72 | 61.04 | 0.48 | 18.36 | 44.61 | 1.66 | 17.55 | 60.80 | 0.70 | 17.08 | 60.42 | 0.75 |
| DecTM (Wu et al., 2021) | 16.60 | 61.15 | 0.59 | 16.48 | 61.33 | 0.49 | 16.50 | 61.27 | 0.55 | 17.29 | 61.09 | 0.65 | 17.22 | 61.10 | 0.73 |
| DVAE (Burkhardt & Kramer, 2019) | 17.12 | 1.00 | **3.78** | 16.26 | 0.64 | **3.74** | 16.94 | 2.67 | 3.07 | 16.55 | 6.61 | 3.11 | 16.78 | 4.79 | 3.06 |
| ETM (Dieng et al., 2020) | 23.48 | 28.14 | 2.94 | 22.52 | 24.95 | 2.99 | 21.59 | 19.53 | 3.03 | 21.54 | 18.13 | 3.08 | 20.78 | 13.48 | 3.18 |
| FASTopic (Wu et al., 2024b) | 14.00 | 60.97 | 0.00 | 36.16 | 49.18 | 3.74 | 31.89 | 39.52 | 3.47 | 33.36 | 40.76 | 3.45 | 15.04 | 61.01 | 0.00 |
| NSTM (Zhao et al., 2021) | 17.76 | 6.66 | 3.58 | 17.50 | 2.98 | 3.65 | 17.44 | 2.09 | **3.71** | 15.83 | 1.57 | 3.52 | 14.89 | 4.04 | 3.15 |
| TSCTM (Wu et al., 2022) | 44.36 | 63.29 | 3.78 | 39.16 | 56.92 | 3.65 | 35.40 | 50.09 | 3.64 | 32.89 | 38.11 | 3.59 | 22.67 | 17.25 | **3.67** |
| SAE-TM (ours) | **55.40** | **70.57** | 3.72 | **54.44** | **75.58** | 3.68 | **45.88** | **70.39** | 3.67 | **41.87** | **66.69** | 3.65 | **38.67** | **64.66** | 3.62 |

*Table 8.* Full results (expanded version of Table 2) for text datasets.

| Num. Topics | 50 | | | 100 | | | 200 | | | 300 | | | 500 | | |
|---|---|---|---|---|---|---|---|---|---|---|---|---|---|---|---|
| | $C_I \uparrow$ | $C_R \uparrow$ | $D \uparrow$ | $C_I \uparrow$ | $C_R \uparrow$ | $D \uparrow$ | $C_I \uparrow$ | $C_R \uparrow$ | $D \uparrow$ | $C_I \uparrow$ | $C_R \uparrow$ | $D \uparrow$ | $C_I \uparrow$ | $C_R \uparrow$ | $D \uparrow$ |
| **CIFAR-100** | | | | | | | | | | | | | | | |
| AVITM (Srivastava & Sutton, 2017) | 33.28 | 76.77 | 3.43 | 33.64 | 78.06 | 3.40 | 30.94 | 77.33 | 3.36 | 30.90 | 75.96 | 3.36 | 29.86 | 73.80 | 3.36 |
| CombinedTM (Bianchi et al., 2021a) | 45.64 | 78.18 | 3.80 | 43.50 | 79.34 | 3.77 | 16.12 | 8.93 | 3.83 | 16.16 | 8.25 | 3.86 | 16.36 | 7.48 | 3.87 |
| DecTM (Wu et al., 2021) | 41.40 | 71.98 | 3.96 | 40.80 | 61.64 | 4.07 | 36.76 | 57.34 | 4.03 | 16.23 | 6.20 | 3.89 | 16.69 | 5.56 | 3.92 |
| DVAE (Burkhardt & Kramer, 2019) | 15.96 | 2.42 | 3.86 | 15.94 | 2.87 | 3.56 | 16.05 | 5.98 | 3.22 | 16.07 | 4.82 | 3.18 | 15.99 | 5.02 | 3.10 |
| ETM (Dieng et al., 2020) | 20.28 | 37.21 | 3.44 | 19.74 | 29.15 | 3.52 | 19.00 | 23.14 | 3.62 | 18.57 | 20.20 | 3.65 | 17.93 | 14.21 | 3.71 |
| FASTopic (Wu et al., 2024b) | 38.36 | 75.50 | 3.52 | 32.62 | 75.10 | 3.40 | 32.13 | 71.67 | 3.49 | 33.15 | 73.13 | 3.45 | 32.60 | 71.95 | 3.49 |
| NSTM (Zhao et al., 2021) | 20.36 | 66.31 | 2.79 | 15.40 | 62.92 | 2.65 | 17.90 | 68.50 | 2.74 | 17.66 | 67.24 | 2.61 | 17.70 | 70.10 | 2.47 |
| TSCTM (Wu et al., 2022) | 46.04 | 83.50 | 3.98 | 43.88 | 83.23 | 3.89 | 40.94 | 80.32 | 3.87 | 36.37 | 68.51 | 3.86 | 28.57 | 44.79 | 3.86 |
| SAE-TM (ours) | 43.16 | 82.52 | 3.51 | 40.18 | 83.05 | 3.50 | 38.99 | 83.15 | 3.49 | 38.56 | 82.25 | 3.48 | 37.65 | 82.02 | 3.50 |
| **FOOD-101** | | | | | | | | | | | | | | | |
| AVITM (Srivastava & Sutton, 2017) | 27.16 | 77.73 | 3.47 | 25.14 | 75.30 | 3.37 | 24.13 | 73.47 | 3.33 | 23.60 | 72.53 | 3.32 | 23.10 | 71.09 | 3.35 |
| CombinedTM (Bianchi et al., 2021a) | 35.68 | 76.15 | 3.77 | 32.06 | 72.56 | 3.74 | 16.08 | 30.87 | 3.61 | 16.45 | 27.71 | 3.63 | 31.08 | 68.38 | 3.75 |
| DecTM (Wu et al., 2021) | 27.48 | 62.72 | 3.86 | 26.04 | 56.41 | 3.85 | 26.00 | 57.55 | 3.83 | 15.41 | 22.99 | 3.64 | 15.41 | 16.34 | 3.71 |
| DVAE (Burkhardt & Kramer, 2019) | 17.84 | 2.52 | 3.66 | 15.86 | 6.58 | 3.61 | 17.11 | 13.44 | 3.12 | 16.38 | 15.12 | 3.11 | 16.84 | 10.61 | 2.92 |
| ETM (Dieng et al., 2020) | 20.52 | 50.49 | 3.45 | 18.96 | 41.18 | 3.52 | 19.38 | 33.55 | 3.56 | 18.58 | 30.08 | 3.60 | 18.12 | 24.96 | 3.65 |
| FASTopic (Wu et al., 2024b) | 32.04 | 64.32 | 3.55 | 30.86 | 64.93 | 3.58 | 29.34 | 62.74 | 3.63 | 28.13 | 60.34 | 3.62 | 28.42 | 59.77 | 3.61 |
| NSTM (Zhao et al., 2021) | 17.32 | 64.66 | 2.64 | 16.76 | 63.80 | 2.54 | 18.43 | 64.57 | 2.43 | 16.55 | 64.04 | 2.43 | 17.52 | 65.78 | 2.70 |
| TSCTM (Wu et al., 2022) | 31.92 | 72.46 | 3.82 | 30.94 | 74.10 | 3.75 | 27.43 | 58.92 | 3.76 | 24.05 | 44.02 | 3.77 | 21.30 | 35.00 | 3.76 |
| SAE-TM (ours) | 34.12 | 84.60 | 3.46 | 33.40 | 83.80 | 3.48 | 32.05 | 83.24 | 3.48 | 30.67 | 82.96 | 3.47 | 29.75 | 81.43 | 3.46 |
| **SUN397** | | | | | | | | | | | | | | | |
| AVITM (Srivastava & Sutton, 2017) | 34.92 | 84.03 | 3.44 | 34.46 | 83.59 | 3.43 | 31.81 | 81.60 | 3.38 | 31.43 | 79.86 | 3.37 | 31.50 | 78.41 | 3.39 |
| CombinedTM (Bianchi et al., 2021a) | 48.92 | 79.42 | 3.95 | 48.12 | 79.84 | 3.91 | 44.36 | 77.52 | 3.88 | 37.35 | 71.97 | 3.97 | 17.11 | 23.13 | 3.71 |
| DecTM (Wu et al., 2021) | 36.92 | 73.08 | 3.95 | 39.32 | 80.85 | 3.85 | 33.76 | 61.64 | 4.02 | 16.71 | 30.38 | 3.58 | 16.12 | 19.32 | 3.72 |
| DVAE (Burkhardt & Kramer, 2019) | 16.96 | 3.44 | 3.93 | 16.20 | 2.86 | 3.73 | 16.79 | 6.30 | 3.19 | 16.89 | 7.96 | 3.08 | 15.74 | 7.06 | 3.09 |
| ETM (Dieng et al., 2020) | 21.20 | 48.91 | 3.37 | 20.38 | 43.76 | 3.46 | 21.01 | 35.90 | 3.53 | 20.31 | 29.81 | 3.57 | 19.29 | 25.45 | 3.63 |
| FASTopic (Wu et al., 2024b) | 37.36 | 72.66 | 3.52 | 39.48 | 74.00 | 3.64 | 39.56 | 73.61 | 3.69 | 38.39 | 73.44 | 3.71 | 38.70 | 74.01 | 3.71 |
| NSTM (Zhao et al., 2021) | 19.16 | 59.15 | 2.85 | 20.80 | 71.25 | 2.93 | 21.36 | 71.74 | 2.92 | 20.13 | 69.62 | 2.86 | 18.92 | 71.10 | 2.83 |
| TSCTM (Wu et al., 2022) | 49.68 | 85.62 | 4.02 | 46.12 | 86.58 | 3.93 | 39.98 | 82.85 | 3.88 | 33.69 | 62.11 | 3.87 | 28.07 | 42.68 | 3.90 |
| SAE-TM (ours) | 50.44 | 88.03 | 3.68 | 46.74 | 90.16 | 3.64 | 44.74 | 90.21 | 3.65 | 43.47 | 89.95 | 3.63 | 42.24 | 89.85 | 3.62 |

*Table 9.* Full results (expanded version of Table 3) for image datasets.

