# OpenReview forum: "Sparse Autoencoders are Topic Models"
_ICML.cc/2026/Conference — ICML 2026 regular_

### Official Review · Reviewer_M1Zc · 2026-02-25

**Soundness:** 1
**Presentation:** 2
**Significance:** 2
**Originality:** 3
**Overall Recommendation:** 2
**Confidence:** 3

**Summary:**

The paper proposes to learn topic models by inferring a document's topic distribution based on the feature activations of sparse autoencoders (SAEs). Word distributions for each SAE topic are obtained by optimizing the bag-of-words likelihood of a corpus. Because the number of topics K in a topic model is typically a lot smaller than the number of SAE latents, SAE features are first clustered into K topics.

As a theoretical justification, the authors also claim that SAEs can be understood as topic models and present a theoretical connection between SAEs and continuous topic models.

They evaluate the approach empirically with intrinsic evaluations (measuring topic coherence and topic diversity) and by extracting topical insights on text and image datasets.

**Compliance With Llm Reviewing Policy:**

Affirmed.

**Final Justification:**

I believe that inferring document-topic distributions in LDA-style topic models from clustered SAE latents (Section 4) is an interesting idea, which is demonstrated empirically by improved topic coherence and diversity and by extracting topical insights on text and image datasets (Section 5).

However, after carefully considering the authors' rebuttal and final comment, and after reading Section 3 again, my key concerns remain. It is my assessment that Section 3 does not prove a meaningful relation between topic models (in the broad sense) and SAEs, and that it is disconnected from the method that the authors propose in Section 4.

As such, claims that are central to the paper are not supported by theoretical or empirical evidence. For instance the title, or the second and third sentence in the abstract:

> We extend Latent Dirichlet Allocation (LDA) to embedding spaces and derive the SAE objective as a maximum a posteriori estimator under this model. This view implies SAE features are thematic components rather than steerable directions.

The rebuttal and comment did not change my assessment because the authors:
1. did not seem receptive to my suggestion to weaken their claims
2. did not provide satisfactory explanations on why their assumptions in Section 3 ($\Sigma_k \rightarrow 0$ and topic independence) are sensible. I maintain that in the limit $\Sigma_k \rightarrow 0$ , CTMs no longer correspond to meaningful topic models.

With respect to point 2, I do not understand the authors' argument in their last comment:
>  In Sec. 3, the ​$\mu_k$ are topic directions in document-embedding space, not embeddings of vocabulary items. Therefore, the reviewer’s phrase “all word vectors associated with a topic are identical” is wrong and the statement “my claim is that the topic-word distributions become degenerate” does not apply, because embeddings of vocabulary items (word vectors) do not appear in Sec. 3 at all and the proposed CTM does not include topic-word distributions.

I agree with the first sentence, which is consistent with my remarks, but the authors then seem to suggest that Section 3 is not about word vectors. Regardless of whether the authors believe word vector is an appropriate terms for $w_n$ (I believe it is), $w_n$'s are presented as CTM's analogue to words. For instance, line 130 right column:

>The N contributions correspond to the number of words in a document, meaning each continuous contribution represents one “embedding word”.

I therefore do not understand how their comment addresses my concern that **if you make the distributions used in Step 3b in Table 1 degenerate ( $\Sigma_k \rightarrow 0$) all $w_n$ linked to the same topic $z_n$ will be identical.** This assumption is not representative for any topic model that I know, and I do not see how any model that operates under this assumption would yield useful insights about document collections.

**Key Questions For Authors:**

- The SAE-TM relies on prior knowledge of the SAE and the underlying LLM. How much of the performance gain is associated to this? For the baselines that also leverage pretrained models (e.g., FastTopic), did you use the same models?

- What precise relation does Section 3.3 prove and under which assumptions?  In Section 3.3 you rely on \Sigma_k -> 0. This collapses each topic distribution to a degenerate distribution. Are you still proving a statement about a topic model in this limit?

- What is the relation between Section 3 and 4? CTM is a generation model for embedding vectors, whereas Section 4 introduce an LDA-style topic model (a generative model of documents and words).

**Limitations:**

For the theoretical analysis the authors should be more transparant about what they prove and under what assumptions, and motivate why these assumptions are sensible.

**Strengths And Weaknesses:**

**Strengths**:
- Experiments across different datasets and modalities.
- The paper presents a new method to fit topic models, which can efficiently vary the number of topics and improves over existing topic models w.r.t. intrinsic metrics (coherence and diversity).

**Weaknesses**:
- [Soundness] The paper puts a lot of emphasis on the fact that SAEs are topic models. However, I believe the theoretical justification for this is flawed. To show SAEs are topic models the authors prove a relation between optimizing SAEs with L1 loss and the optimization of the continuous topic model (CTM) they introduce (Table 1). However:
    * Contrary to what the authors suggest, a CTM isn't an extension to LDA because it doesn't associate topics with discrete units such as words. It is a generative model for continuous vectors.
    * The theoretical analysis (Section 3.3) only holds for the trivial case of a CTM where all topic distribution collapses to degenerate distributions, where all samples drawn from it are identical (see step 3 in Table 1b and assumption \Sigma_k -> 0 on line 202).
    * The analysis also assumes independence across topics (line 193), which doesn't hold for more recent SAE variants (e.g., Matryoshka SAEs).
- [Soundness] The theoretical analysis in Section 3 is disconnected from the method that is proposed in Section 4 and evaluated in Section 5. The SAE-based topic model that is introduced in Section 4 is unrelated to the CTM and the assumptions that were made in Section 3.3. For instance, in Section 4 topics are associated with (non-trivial) word distributions, and it clusters SAE dimensions before using them as topics.
- [Presentation] The theoretical analysis could be presented more clearly:
  - The wording of the assumptions in the theoretical analysis (Section 3.3) is quite ambiguous at the moment: "By independence across topics line 193; "corresponding to the limit \Sigma_k -> 0" line 202) It would be more transparent to clearly state these as assumptions.
- [Presentation] The baselines and their experimental setup could be described more clearly. To my understanding at least one of the baselines relies on a foundation model to compute embeddings but this is not described.
- [Significance] It's difficult to assess where the improvement of SAE-TM comes from: An inherent effectiveness of SAE-informed topics; or the prior knowledge that is introduce by the embedding models/ the datasets on which the SAEs were trained. It seems the latter could also be leveraged by baselines (e.g., FastTopic). Are the same foundation models used for SAE-TM and the baselines?

Overall, I think the paper would be more sound, easier to read, and more consistent if it were less ambitious and more transparent in its theoretical claims and analysis.

---

> ### Author Rebuttal · Authors · 2026-03-29
>
> We thank the reviewer for the detailed feedback and for recognizing our contributions. We believe the main concerns rest on misunderstandings of Sec. 3 and its connection to Sec. 4, which we hope to address below.
>
> ---
>
> > Contrary to what the authors suggest, a CTM isn't an extension to LDA because it doesn't associate topics with discrete units such as words. It is a generative model for continuous vectors.
>
> We think this concerns the precise definition of "topic model". Wikipedia defines it as "a probabilistic, neural, or algebraic model for discovering the abstract topics that occur in a collection of documents". A topic model (TM) is about the latent topics it discovers, and word emissions are one possible interpretation of the topics, not a definitional requirement. Hence, CTM is a TM, and we show that SAEs perform MAP inference under it.
>
> We agree the CTM isn't identical to LDA, but it’s a useful embedding-space analogue. It preserves the Dirichlet prior and per-observation topic sampling, but replaces discrete word emissions with continuous directions. This paradigm is well-established. Dieng et al. [1] note: "one common strategy is to convert the discrete text into continuous observations of embeddings, and then adapt LDA to generate real-valued data". Such models, like "Gaussian LDA" [2], are accepted as TMs without qualification. CTM follows this paradigm. We are happy to describe the CTM as "inspired by" rather than "extending" LDA to avoid ambiguity.
>
> [1] Dieng et al: Topic modeling in embedding spaces, TACL 2020\
> [2] Das et al: Gaussian LDA for topic models with word embeddings, ACL 2015
>
> ---
>
> > You rely on \Sigma_k -> 0. Are you still proving a statement about a topic model in this limit?
>
> We are happy to clarify, as we believe this is a misreading of the role of  $\Sigma_k$:
> * The function of $\Sigma_k$ is to govern within-topic directional spread of individual contributions ($w_n \sim N(\mu_k, \Sigma_k)$ where $k = z_n$), not the document-level topic mixture
> * The effect of $\Sigma_k \to 0$ is that individual contributions align directly with the topic's mean direction ($\mu_k$)
> * But the mixture structure doesn’t collapse. It remains fully non-degenerate:
>    - The document-level topic proportions remain stochastic: $\theta \sim Dir(\alpha)$
>    - The scale parameter remains stochastic: $s \sim Ga(\kappa, \beta)$
>
> The assumption simply matches standard SAEs, which use deterministic decoders, but doesn’t make our contribution trivial.
>
> ---
>
> > Analysis also assumes independence across topics which doesn't hold for more recent SAE variants (e.g., Matryoshka SAEs).
>
> Our analysis applies to L1/JumpReLU SAEs (Sec. 3.3) and TopK SAEs (App. A), i.e. the most widely used variants and the ones in all our experiments. We see this as an opportunity: Matryoshka SAEs implement hierarchical structure that could model HDPs (also see Reviewer 9u7R's suggestion about HDPs). This is promising future work building on our foundation.
>
> ---
>
> > It would be more transparent to clearly state these as assumptions.
>
> In the revised version, we directly state these assumptions in the first paragraph of Sec. 3.3, i.e. starting from line 159.
>
> ---
>
> > What is the relation between Section 3 and 4?
>
> Sec. 3 motivates why we use SAEs as topic models: they optimize the MAP of the CTM. This directly connects to Sec. 4, where we outline how SAEs apply in practice for topic modeling. Sec. 3 covers theory, and Sec. 4 covers practice. To be clear, merging and interpretation steps in Sec. 4 build on a frozen SAE exactly as outlined in Sec. 3, and don't propose a new model. Topic merging enables the efficient creation of TMs with a varying number of topics from SAEs, a strength explicitly recognized by the reviewer. Association of SAE features with word distributions happens mainly because we evaluate SAE-TMs on the same terms as other TMs, but SAE-TMs allow other interpretation methods, see our response to reviewer BYLg. Our updated version will make the connection between these sections tighter.
>
> ---
>
> > How much of the performance gain is associated using pretrained embeddings? For the baselines that also leverage pretrained models (e.g., FastTopic), did you use the same models?
>
> All models operate on the same document embeddings, if applicable, i.e. we use Granite-R2 embeddings for SAEs and baselines (CombinedTM, Fastopic). Thus, the advantage isn’t due to better embeddings. Word emission probabilities are also always learned on the same data. We think the main advantage of SAE-TMs is decoupling the actual topic modeling (SAE feature learning) from their interpretation (learning the word emission probabilities). This gives more flexibility to induce better topics by not having to learn both topics + their interpretation at the same time.
>
> ---
>
> We hope we have addressed all points that were unclear. If anything else remains, we hope to receive a response to our rebuttal. Otherwise, we thank the reviewer for increasing their score.

---

> > ### Author Rebuttal · Reviewer_M1Zc · 2026-04-02
> >
> > I have read the authors' rebuttal. It did not alleviate my main concerns, however. The authors' clarifications are in line with my initial understanding. I will clarify my position:
> >
> > * I agree that CTMs fit the definition of a topic model, my comment was that they are not an extension of LDA. Describing CTMs as "inspired by" LDA would indeed be a more appropriate characterization, but it does not address the fact that Sect. 3 is proving a property about a different type of model (CTM) than what the authors actually propose and evaluate (an LDA variant).
> > * `"Hence, CTM is a TM, and we show that SAEs perform MAP inference under it."`
> >   This is not what the authors prove because their derivation only holds in the limit where the covariance matrices of the topic-word distributions go to zero  ($\Sigma_k \rightarrow 0$). This does not hold in general for CTMs. In this limit, the *topic-word vector distributions* (i.e., the distributions from which you sample the word vectors) become degenerate: all vectors that are drawn from a topic-word vector distribution are identical. The authors therefore only prove that SAEs perform MAP inference under a special case of CTMs that collapses a crucial aspect of topic models.
> > * To my understanding, the assumption further implies that: 1) all word vectors associated with a topic are identical, save for their magnitudes and an error term $\epsilon$ ; 2) a word vector can only be associated with one topic (unless $\mu_k$ of different topics are close relative to the error term $\epsilon$). These assumptions are not sensible to me and do not match the method the authors later propose and evaluate.
> > * `"We are happy to clarify, as we believe this is a misreading of the role of [...]"`
> > 	My understanding matches this description, but as explained in the previous bullet, my claim is that the topic-word distributions become degenerate, not the document-topic distributions.
> > * `"The assumption simply matches standard SAEs"` I believe this argument is circular, because the authors' proof aims to reduce CTM MAP inference to training SAEs with L1 loss. Therefore the assumptions need to be sensible for CTMs not just for SAEs.
> >
> > In sum, my assessment remains that 1) the authors did not prove that SAEs are topic models, and 2) the theoretical analysis is disconnected from the method that was actually proposed and evaluated.

---

> > > ### Author Response · Authors · 2026-04-05
> > >
> > > We thank the reviewer for the detailed clarification.
> > >
> > > Indeed we completely agree with your observations that:
> > > 1. “CTMs fit the definition of a topic model” and
> > > 2. “the authors [...] prove that SAEs perform MAP inference under a special case of CTMs”
> > >
> > > However, there are two serious mistakes in the reviewer’s statement:
> > >
> > > In Sec. 3, the $\mu_k$​ are topic directions in document-embedding space, not embeddings of vocabulary items. Therefore, the reviewer’s phrase “all word vectors associated with a topic are identical” is wrong and the statement “my claim is that the topic-word distributions become degenerate” does not apply, because embeddings of vocabulary items (word vectors) do not appear in Sec. 3 at all and the proposed CTM does not include topic-word distributions.
> > >
> > > In Sec. 4 we in fact do not describe the exact generative model analyzed in Sec. 3. It introduces additional layers beyond the CTM introduced in Sec. 3. In the narrower and perhaps more common NLP usage that the reviewer refers to, a topic model includes a word-level representation of each topic. In the broader latent-variable sense used in our paper, the topic model is the mechanism that assigns each document a sparse mixture of shared latent topics. In this context, Reviewer 9u7R found our perspective “an extremely interesting, clear-headed & refreshing take”. Under this definition, the SAE is the topic model in both sections. Sec. 4 then adds a post hoc interpretation that maps these latent topics to word distributions so they can be inspected and evaluated with standard topic model metrics. We will state this explicitly in the paper and clarify Sec. 4 accordingly.
> > >
> > > In summary, Sec. 3 clarifies under which specific conditions the SAE objective corresponds to MAP inference in a topic model, motivating SAEs applied as topic models for the first time in the literature. We do not claim that SAEs are equivalent to the full CTM or classical word-generating topic models in their full generality. Sec. 4 embeds SAEs, which are the target of the analysis in Sec. 3, into an applicable method that (1) retains the SAE as the latent topic-assignment method identified in Sec. 3; (2) allows its evaluation through standard metrics, such as coherence; and (3) allows fair comparison to standard baselines by expressing the learned latent topics through a similar method to learn word distributions as is used in other topic models.
> > >
> > > We hope that these clarifications help resolve the reviewer’s doubts and they appreciate our fresh perspective on SAEs and topic modeling as pointed out by reviewers 9u7R (“actually novel and useful”), BYLg (“elegant derivation”), and FeWL (“refreshing lens”).

---

### Official Review · Reviewer_FeWL · 2026-03-12

**Soundness:** 3
**Presentation:** 2
**Significance:** 3
**Originality:** 3
**Overall Recommendation:** 4
**Confidence:** 4

**Summary:**

This paper presents a broad context regarding the interpretability of embeddings, specifically proposing that Sparse Autoencoders (SAEs) can be theoretically and practically grounded as topic models. A core challenge analyzed by this paper is how to bridge the gap between the mechanistic interpretability of SAEs (often viewed as steerable features) and the semantic organization of probabilistic topic models. The authors attempt to formalize this connection by extending Latent Dirichlet Allocation (LDA) to continuous embedding spaces and deriving the SAE objective as a Maximum A Posteriori (MAP) estimator. They introduce the "SAE-TM" framework, which uses pre-trained SAEs to extract "topic atoms" that are subsequently merged into topics. The method is evaluated on text and image datasets against various neural topic modeling baselines.

**Compliance With Llm Reviewing Policy:**

Affirmed.

**Final Justification:**

The new results, especially those addressing the fairness concern by training SAE-TM on downstream datasets, are a useful addition and resolve an important issue in my original review. They make it clearer that the reported gains are not simply an artifact of large-scale pretraining. Based on this, I am updating my scores to:

Soundness: 3
Overall Recommendation: 4

**Key Questions For Authors:**

- Fairness & Pre-training: How can you justify the fairness of comparing SAE-TM (pre-trained on hundreds of millions of samples) against baselines trained only on small downstream datasets? Have you considered using off-the-shelf general-purpose SAEs to demonstrate transferability, or conversely, training the SAE only on the downstream data to strictly evaluate the modeling architecture?
- Corpus Specificity: High coherence is expected from large language models. Can you provide results on Corpus Coverage or representativeness? We need to ensure the model isn't just retrieving generic themes from its pre-training memory that are irrelevant to the specific document collection.
- Methodological Identity: Is the proposed method fundamentally different from simply using an SAE as a frozen encoder within a standard VAE or clustering framework? If so, where does the distinct modeling advantage lie?
- Theoretical Role: Does the complex LDA-to-SAE derivation provide any actual constraints used in the loss function during training or merging? If not, would the empirical results change if the theory were removed and the paper simply framed as "clustering sparse features"?
- Pivot Suggestion: You mention that "SAE features should be seen as thematic clusters... rather than a monosemantic mechanism." This is a compelling insight. Would you consider reframing the paper to focus on this analysis of the nature of SAEs, rather than framing it as a state-of-the-art topic model competition (which is currently flawed due to data scale disparities)?

**Limitations:**

The authors have not adequately discussed the computational costs and environmental impact of training massive SAEs compared to lightweight topic models. They also barely touch upon the handling of non-thematic structures in embeddings, which is a critical limitation for topic modeling.

**Strengths And Weaknesses:**

Strengths:
- Intuitive Perspective: The proposal to view SAE features as "topic atoms" that combine to explain embeddings is a refreshing lens compared to the standard "monosemanticity" view.
- Multimodal Potential: The application to both text and images (e.g., the analysis of Ukiyo-e prints) demonstrates the potential versatility of embedding-based approaches.

Weaknesses:
1. Critical Flaws in Experimental Soundness (The experimental methodology is fundamentally flawed, which is the primary reason for the low soundness score):
- Unfair Comparison: The proposed SAE-TM relies on SAEs pre-trained on massive datasets (480 million text sections, 360 million images). In contrast, the baseline models (such as CombinedTM and FASTopic) are typically trained from scratch on small downstream datasets (e.g., News-20K with only ~18k documents). This is not an "apples-to-apples" comparison. The high coherence scores are likely a result of the massive pre-training capturing general language/image statistics, rather than the efficacy of the proposed topic modeling framework itself.
- Insufficient Metrics: The evaluation relies heavily on Coherence and Diversity. However, with strong pre-trained encoders, it is easy to retrieve generic, highly coherent words (e.g., "one, two, three") that have little to do with the specific thematic structure of the target corpus. The paper lacks Coverage or Representativeness metrics (e.g., NPMI calculated specifically on the downstream corpus). Furthermore, averaging results across datasets hides dataset-specific failures or nuances.

2. Presentation and Theoretical Obfuscation:
- Unnecessary Complexity: The paper dedicates significant space to a complex derivation connecting LDA to "Continuous Topic Models." This heavy mathematical formalism obscures the actual, straightforward mechanism: using an SAE as a frozen encoder within a clustering framework. The derivation feels like a post-hoc justification rather than a functional part of the loss function or training constraints.
- Missing Details: Critical details are relegated to the Appendix (e.g., the specific topic merging algorithm in Appendix C). Furthermore, the term "Feature-Word Association" mentioned in Figure 2 is neither defined nor does it appear in the main text, causing confusion.
- Missing Citations: The paper overlooks relevant work in contextual and graph-based topic modeling. Specifically, it misses *CEMTM: Contextual Embedding-based Multimodal Topic Modeling*.

3. Methodological Ambiguity:
- Lack of Novelty in Method: Structurally, the method appears to be a VAE-like framework using an SAE as a frozen encoder. The novelty seems limited to the application of SAEs to this task rather than a new modeling technique.
- Missing Technical Details: The authors acknowledge that "SAE feature interpretation can be improved" and that "activation strengths do not always align with topic importance," but provide no details on how these issues are handled. Details regarding training efficiency and computational resources are also missing.

---

> ### Author Rebuttal · Authors · 2026-03-29
>
> We thank the reviewer for the extensive review, highlighting our “intuitive perspective” and the “multimodal potential” of our method. Below, we provide details to address the reviewer’s concerns:
>
> ---
>
> > Fairness & Pre-training
>
> We would like to clarify a potential misunderstanding: We originally wanted to emphasize reusability of SAE-TMs, in which case the comparison isn’t unfair because we can use the same SAE across multiple downstream datasets and topic numbers.
>
> We now train SAE-TMs only on the individual datasets (e.g. only the 18K-sample news-20k dataset). See our response to BYLg for the new, **improved** scores. They are **stronger** (e.g. $C_I$ avg. 48.17% new vs. 45.57% old) than original scores in the paper, because now the SAE is fine-tuned to downstream data. For pretrained SAEs, this motivates fine-tuning as a promising option.
>
> We would also like to highlight that SAE-TMs are very efficient. Training on 50M embeddings (e.g. 100 epochs, 3136 features, Twitter dataset) takes 10 min (< 15GB VRAM), topic interpretation takes 15 min. This is much less than most baselines (e.g. 60 min for DecTM, 100 topics, 100 epochs, Twitter dataset), especially given that SAEs scale better to higher numbers of topics (emphasized by 9u7R). We add details on computational cost to the updated version.
>
> ---
>
> > Corpus Specificity: High coherence is expected from large language models / Insufficient Metrics
>
> We show that SAE-TM topics aren't only coherent but also relevant: Given a document and assigned topics, we sample one active topic and one inactive topic and let an LLM decide if the topic is relevant to the document. If topics are not relevant, the LLM cannot distinguish between active and inactive topics. SAE-TMs **outperform baselines** as the next table shows (SAE-TM and best baseline avg. scores across text datasets):
>
> | | 50 | 100 | 200 | 300 | 500 | Mean |
> |-|-|-|-|-|-|-|
> | SAE-TM | 0.64 | 0.64 | 0.66 | 0.68 | 0.69 | 0.67 |
> | AVITM | 0.61 | 0.62 | 0.63 | 0.64 | 0.60 | 0.62 |
>
> Accuracy on irrelevant topics is close to 100%, but most topics are broad, so many active topics are classified as irrelevant. The margin on active topics is particularly favorable for SAE-TM, avg. 38% vs. 29%. Full results are here: https://anonymous.4open.science/r/icml-17645-figures-56DD/
>
> Finally, we add tables with individual dataset results to the updated version, but do not find any dataset-specific trends.
>
> ---
>
> > Presentation and Theoretical Obfuscation / Theoretical Role
>
> **Unnecessary Complexity**: The derivation in Sec. 3 is important to motivate why SAEs can be used for topic modeling and show what SAEs actually learn. We and other reviewers (BYLg, 9u7R) find this very interesting, e.g. BYLg highlights the “elegant” derivation and 9u7R finds this an “extremely interesting, clear-headed & refreshing take”. Sec. 4 (lines 177 to 265) then applies SAE topic models in a practical method. Please also see our response to M1Zc for an explanation of how Sec. 3 and Sec. 4 connect.
>
> **Missing Details/Citations**: App. C offers an extended description of topic merging, but all relevant details are in lines 220 to 250. We will stress better that App. C only contains a longer description for the reader’s convenience, and rephrase “Feature-Word Association”. Also, we will add the CEMTM paper to the Related Work.
>
> ---
>
> > Methodological Ambiguity / Methodological Identity
>
> **Lack of Novelty in Method**: The main novelties are
> * Using SAEs for Topic Modeling
> * Ability to efficiently change the number of topics via merging
> * SAE-TMs operate entirely in latent space (i.e. reconstruct the input embedding, not words)
> * Flexibility in the topic interpretation method, see our response to reviewer BYLg
> * Greater interpretability: Through fine-grained topic atoms we can explain why a coarse topic was assigned which isn't available when methods only assign coarse topics
>
> We will stress these advantages, which are also technical novelties.
>
> **Missing Technical Details**:
> * "SAE feature interpretation can be improved" means the possibility of interpreting SAE topics differently than other TMs. We don’t explore this here to fairly compare to baselines, see our response to BYLg for other interpretation methods
> * "Non-thematic structures" means the general property of embeddings where they represent non-topic-related info, like sentiment or document length. This applies to all embedding-based TMs, including established ones like CombinedTM
>
> We mention these in lines 429 to 433 and will add details.
>
> ---
>
> In this rebuttal, we have shown that a data-matched comparison also favors SAE-TMs, topics are not only coherent but also relevant, and we have clarified our theoretical contribution and novelty. We believe the reviewer’s initial score was based on a misunderstanding that we hope to have addressed in our rebuttal.
>
> If anything else remains, we hope to receive a response to our rebuttal. Otherwise, we thank the reviewer for increasing their score.

---

> > ### Author Rebuttal · Reviewer_FeWL · 2026-04-02
> >
> > The new results, especially those addressing the fairness concern by training SAE-TM on downstream datasets, are a useful addition and resolve an important issue in my original review. They make it clearer that the reported gains are not simply an artifact of large-scale pretraining.

---

> > > ### Author Response · Authors · 2026-04-05
> > >
> > > Thank you very much for the response and your support of our paper! We are happy that our new experiments could fully resolve your initial concerns.

---

### Official Review · Reviewer_BYLg · 2026-03-13

**Soundness:** 3
**Presentation:** 4
**Significance:** 3
**Originality:** 3
**Overall Recommendation:** 5
**Confidence:** 2

**Summary:**

The authors propose a framework called SAE-TM that applies Sparse Autoencoders (SAEs) for a topic modeling task. They connect Latent Dirichlet Allocation (LDA) to continuous embedding spaces by introducing a Continuous Topic Model (CTM). They prove mathematically that finding the maximum a posteriori (MAP) estimator for the CTM relates to the standard SAE training objective. The proposed framework extracts human-readable topics from frozen SAEs by learning a word emission matrix, projecting these into dense semantic space using pre-trained word embeddings, and merging the feature dictionary down to a small number of topics using k-means clustering. Because the method uses embeddings, it can be applied to both text and image domains, though interpreting image features still heavily relies on generated text captions.

**Compliance With Llm Reviewing Policy:**

Affirmed.

**Final Justification:**

All my concerns have been resolved. I keep my positive score.

**Key Questions For Authors:**

N/A

**Limitations:**

yes

**Strengths And Weaknesses:**

Strengths:
- The mathematical derivation connecting the CTM's MAP estimator to the SAE loss function is elegant. The authors demonstrate that SAE features are thematic mixtures.
- The authors demonstrate practical utility by comparing the thematic composition of different large-scale image datasets.
- The case study tracking evolving themes in Japanese woodblock prints is excellent. The authors apply their pretrained SAE to text embeddings of generated captions to capture the thematic changes of the artwork.

Weaknesses:
- The theoretical elegance of the paper relies heavily on deriving the $L_1$ penalty from the CTM. However, the actual experiments utilize a BatchTopK SAE, bypassing the $L_1$ penalty entirely. While the authors provide an appendix section relating fixed-sparsity to a hard deterministic support constraint, the main text has to be self-contained and consistent with experiments.
- Although the model processes image embeddings, interpreting the resulting features and learning word emission probabilities requires detailed, long-form text captions generated by an external model (INTERNVL3.5-14B). This means the framework is still dependent on textual representation.

Minor Weaknesses
- In Eq. 11, the background weight $\pi$ is hardcoded to 0.3 based on manual validation. It is unclear how sensitive the final topics are to this hyperparameter.

---

> ### Author Rebuttal · Authors · 2026-03-29
>
> We thank the reviewer for the positive evaluation of our work, and we are particularly happy that the reviewer appreciates our analysis on evolving themes in Japanese woodblock prints and large scale investigation of dataset differences. Below, we respond to the remaining concerns mentioned in the review.
>
> ---
>
> > The theoretical elegance of the paper relies heavily on deriving the L1 penalty from the CTM. However, the actual experiments utilize a BatchTopK SAE, bypassing the penalty entirely. While the authors provide an appendix section relating fixed-sparsity to a hard deterministic support constraint, the main text has to be self-contained and consistent with experiments.
>
> For the submission, we followed best practices for SAE training, which suggest using BatchTopK SAEs over L1-regularized SAEs, but we agree that consistency is important. Therefore, we now train L1-SAEs on all individual datasets (this is suggested by Reviewer FeWL). The results **improve** upon what we originally reported in the paper, likely because training on datasets directly makes the topics better aligned. We will update the paper with these new, **better** scores:
>
> | Method | | 50  | | | 100 | | | 200 | | | 300 | | | 500 | |
> |---|---|---|---|---|---|---|---|---|---|---|---|---|---|---|---|
> | | $C_I$ | $C_R$ | $D$ | $C_I$ | $C_R$ | $D$ |  $C_I$ | $C_R$ | $D$ | $C_I$ | $C_R$ | $D$ | $C_I$ | $C_R$ | $D$ |
> | SAE-TM (text)     | 55.69  | 76.3   | 3.70 | 52.45   | 77.3   | 3.67 | 47.32   | 75.3   | 3.63 | 44.24   | 73.8   | 3.61 | 41.13   | 71.0   | 3.58 |
> | SAE-TM (image)    | 42.57  | 85.0   | 3.55 | 40.11   | 85.7   | 3.54 | 38.59   | 85.5   | 3.54 | 37.57   | 85.1   | 3.53 | 36.54   | 84.4   | 3.53 |
>
> ---
>
> > Although the model processes image embeddings, interpreting the resulting features and learning word emission probabilities requires detailed, long-form text captions generated by an external model (INTERNVL3.5-14B). This means the framework is still dependent on textual representation.
>
> We would like to clarify that for image embeddings, we only need text captions to create topic interpretations that resemble baseline methods. In practice, SAE TMs trained on image embeddings are interpretable through more direct methods, such as retrieving the top-$k$ most activating images for an SAE feature and creating a weighted list of keywords from this set by an MLLM, which is a standard method for interpreting vision SAEs. That would also be more efficient than captioning all images, although any natural language interpretation of visual topics requires an MLLM or captioning model as an intermediary. We align with the baselines in this paper to avoid additional confounders and simplify the setup. However, we emphasize that this is not a strict constraint. We will clarify this in the updated version.
>
> ---
>
> > In Eq. 11, the background weight  is hardcoded to 0.3 based on manual validation. It is unclear how sensitive the final topics are to this hyperparameter.
>
> We agree that a systematic ablation of the background weight is interesting. Below we show the results for background weights from 0.1 to 0.9, with the same settings (text datasets, number of topics) as in the paper. We find that scores are generally stable (up to extreme values around 0.9), but the maximum performance is reached for values around 0.2. We will add these results to the paper.
>
> | Init $\pi$ | | 50 | | | 100 | | | 200 | | | 300 | | | 500 | |
> |---|---|---|---|---|---|---|---|---|---|---|---|---|---|---|---|
> | | $C_I$ | $C_R$ | $D$ | $C_I$ | $C_R$ | $D$ | $C_I$ | $C_R$ | $D$ | $C_I$ | $C_R$ | $D$ | $C_I$ | $C_R$ | $D$ |
> | 0.1 | 0.53 | 77.7 | 3.58 | 0.48 | 77.0 | 3.56 | 0.45 | 76.7 | 3.52 | 0.43 | 75.0 | 3.50 | 0.39 | 72.2 | 3.47 |
> | 0.2 | 0.56 | 78.2 | 3.65 | 0.51 | 77.7 | 3.62 | 0.47 | 76.9 | 3.59 | 0.45 | 75.9 | 3.57 | 0.42 | 73.1 | 3.55 |
> | 0.3 | 0.55 | 77.3 | 3.67 | 0.52 | 78.2 | 3.64 | 0.46 | 75.9 | 3.60 | 0.44 | 74.3 | 3.58 | 0.41 | 71.4 | 3.57 |
> | 0.4 | 0.55 | 77.5 | 3.70 | 0.51 | 77.0 | 3.67 | 0.47 | 75.2 | 3.64 | 0.44 | 73.4 | 3.62 | 0.41 | 71.0 | 3.60 |
> | 0.5 | 0.54 | 75.3 | 3.71 | 0.52 | 76.4 | 3.71 | 0.47 | 74.5 | 3.68 | 0.44 | 72.4 | 3.66 | 0.41 | 70.1 | 3.64 |
> | 0.6 | 0.53 | 74.6 | 3.73 | 0.50 | 75.9 | 3.73 | 0.47 | 73.5 | 3.71 | 0.45 | 72.1 | 3.70 | 0.41 | 68.8 | 3.68 |
> | 0.7 | 0.54 | 75.5 | 3.78 | 0.51 | 74.6 | 3.76 | 0.47 | 72.0 | 3.75 | 0.44 | 70.2 | 3.74 | 0.41 | 67.2 | 3.72 |
> | 0.8 | 0.54 | 72.6 | 3.80 | 0.49 | 71.2 | 3.81 | 0.46 | 69.0 | 3.79 | 0.43 | 67.0 | 3.78 | 0.40 | 64.7 | 3.77 |
> | 0.9 | 0.53 | 69.9 | 3.86 | 0.50 | 69.5 | 3.86 | 0.44 | 66.0 | 3.85 | 0.41 | 63.1 | 3.85 | 0.38 | 60.3 | 3.83 |

---

> > ### Author Rebuttal · Reviewer_BYLg · 2026-04-03
> >
> > I thank the authors for their response. I will keep my positive score.

---

> > > ### Author Response · Authors · 2026-04-05
> > >
> > > Thank you very much for your continuing support of our paper!

---

### Official Review · Reviewer_9u7R · 2026-03-13

**Soundness:** 4
**Presentation:** 4
**Significance:** 4
**Originality:** 3
**Overall Recommendation:** 6
**Confidence:** 4

**Summary:**

This paper argues that SAEs can be treated as topic models on embeddings, with SAE features resembling thematic topic atoms - rather than monosemantic steering directions. It derives the SAE loss from a modified form of latent dirichlet allocation on embeddings, and shows that trained SAEs achieve state-of-the-art performance on text only topic modelling benchmarks, suggesting they can be productively used for thematic analysis of embedding spaces.

**Compliance With Llm Reviewing Policy:**

Affirmed.

**Final Justification:**

This is an extremely valuable piece of work for two kinds of practitioners - 1) those using SAEs as interpretability tools, to more rigorously understand what they are optimising and why they have the properties they do beyond thinking of them as "monosemantic steering vectors", and 2) those interested in extracting thematic topics from large scale datasets, using a principled tool with desirable properties such as extracting multiple relevant topics.

**Key Questions For Authors:**

How flexible is this formulation to the chosen generative model - for instance, if I wanted something similar to BERTopic or some other flavour of a DP/HDP, would I be able to derive a modified SAE loss that targets this, or can SAEs not capture that kind of behaviour?

**Limitations:**

yes

**Strengths And Weaknesses:**

This is an extremely interesting, clear-headed & refreshing take on treating SAEs as thematic topic models. It is theoretically grounded, clarifies observed SAE behaviour & offers a valuable and much-needed technique for the community to use (with good empirical performance). The paper's take is actually novel and useful to the field as a powerful technique for a well-defined problem (esp. since many complex topic modelling approaches scale poorly) - and it is both theoretically grounded and well demonstrated empirically.

More of a nice to have than weakness - an illustrative example would help strengthen some of the theoretical intuitions build in the derivation of the SAE loss, in particular how SAEs constrain the behaviour to the high-activity small-contribution limit.

---

> ### Author Rebuttal · Authors · 2026-03-29
>
> We thank the reviewer for the great support of our paper. Below, we add details regarding the suggestions:
>
> > More of a nice to have than weakness - an illustrative example would help strengthen some of the theoretical intuitions build in the derivation of the SAE loss, in particular how SAEs constrain the behaviour to the high-activity small-contribution limit.
>
> We have created a figure to illustrate the theoretical intuitions here: [https://anonymous.4open.science/r/icml-17645-figures-56DD/](https://anonymous.4open.science/r/icml-17645-figures-56DD/).
>
> The figure shows how embeddings for 3 documents are generated under different conditions. On the left, a document embedding is formed by only a few large topic contributions ($N=5$). In this case, possible embeddings form a rigid, clumpy grid. This discrete pattern doesn’t harmonize well with the continuous $L_2$ reconstruction loss used by SAEs. However, on the right, we move to the high-activity, small-contribution limit ($N=500$). There, the clumpy grid smoothes out into Gaussian clouds centered at each document's expected embedding ($W\theta$).
>
> This visual shift explains how we can apply the standard SAE objective (i.e. the continuous $L_2$ loss) to a model based on discrete, word-like topic contributions (the CTM). At the same time, making each individual contribution small prevents the overall embedding size from increasing without bound, which mathematically yields the $L_1$ sparsity regularization.
>
> > How flexible is this formulation to the chosen generative model - for instance, if I wanted something similar to BERTopic or some other flavour of a DP/HDP, would I be able to derive a modified SAE loss that targets this, or can SAEs not capture that kind of behaviour?
>
> One could use SAEs within BERTopic's framework, for example, by taking the highest-scoring topic from the SAE decomposition as the cluster assignment i.e. SAEs become the clustering step. Also, we can discard SAE features that don’t frequently activate, like the unclustered documents in HDBSCAN which was originally proposed for BERTopic. That being said, the main strength of SAEs vs. embedding clustering is that they can discover multiple relevant directions, i.e. topics.
>
> Regarding HDPs, the main difficulty is that the stick-breaking construction allows for an unbounded number of topics, but SAEs have a fixed dictionary size. However, there are in fact hierarchical variants of SAEs:
>
> * Matryoshka SAEs [1] train nested autoencoders at multiple scales and yield a natural feature (or, in our case, topic) hierarchy
> * Matching-Pursuit SAEs [2] explicitly model hierarchical structure over directions, which can also be translated to a topic hierarchy
>
> So SAEs can be used to induce topic hierarchies, but we likely won’t get a formal equivalence of HDPs and SAEs.
>
> [1] Bussmann et al: Learning multi-level features with matryoshka sparse autoencoders, ICML 2025\
> [2] Costa et al: From flat to hierarchical: Extracting sparse representations with matching pursuit, NeurIPS 2025

---

> > ### Author Rebuttal · Reviewer_9u7R · 2026-04-03
> >
> > Thanks to the authors for the detailed response!  the visual intuition is quite useful, and so are the detailed mappings to other techniques - as a practitioner these are the kinds of intuitions that help me make design decisions, worth mentioning in the manuscript if possible (even if in the appendix)

---

> > > ### Author Response · Authors · 2026-04-05
> > >
> > > Thank you very much for the response! We are happy that you find our additions useful and certainly we will add the additional details to the updated version.

---

### Decision · Program_Chairs · 2026-04-30

**Decision:**

Accept (regular)

**Comment:**

This paper shows how SAEs can be viewed as topic models on document embeddings and proposes a framework called SAE-TM that extracts topics from pre-trained SAEs. Reviewers praised the significance of this work, as it provides a “refreshing” take on SAEs and is a valuable perspective for both those who use SAEs for interpretability and those who want to extract topics from large datasets. The paper offers solid theoretical grounding and good empirical performance across modalities.

Two main concerns were raised by reviewers during the rebuttal phase. The first pertained to the fairness of the empirical comparison as the proposed SAE-TM approach relies on large-scale pre-training. The authors addressed this concern by adding new results training SAE-TMs on individual datasets. The second concern centered on issues with the theoretical justification for viewing SAEs as topic models due to the assumption that individual contributions align with an overall mean direction. After reviewing the arguments on both sides, I do not found this assumption to be problematic. First, it is understood that any reinterpretation of method like SAEs will not necessarily exactly correspond with an approach derived directly from the principles of topic modeling. Second, the overall goal of the proposed view is not to obtain per-word embeddings but rather to infer a representation for the document as a whole, decomposed suitably as a mixture of topics. Taken together, the inability for an SAE to directly capture variability at a word level is not a critical flaw.

Overall, due to the significance and soundness, I recommend acceptance.